# APOE Peripheral and Brain Impact: APOE4 Carriers Accelerate Their Alzheimer Continuum and Have a High Risk of Suicide in PM_2.5_ Polluted Cities

**DOI:** 10.3390/biom13060927

**Published:** 2023-05-31

**Authors:** Lilian Calderón-Garcidueñas, Jacqueline Hernández-Luna, Mario Aiello-Mora, Rafael Brito-Aguilar, Pablo A. Evelson, Rodolfo Villarreal-Ríos, Ricardo Torres-Jardón, Alberto Ayala, Partha S. Mukherjee

**Affiliations:** 1College of Health, The University of Montana, Missoula, MT 59812, USA; 2Universidad del Valle de México, Mexico City 14370, Mexico; rafael.brito@uvmnet.edu; 3Radiology Department, HMG, Mexico City 04380, Mexico; radiologa.jacqueline.hernandez@gmail.com; 4Otorrinolaryngology Department, Instituto Nacional de Cardiología, Mexico City 14080, Mexico; aiellov@hotmail.com; 5Facultad de Farmacia y Bioquímica, Universidad de Buenos Aires, Buenos Aires C1113 AAD, Argentina; pevelson@ffyb.uba.ar; 6Universidad Autónoma de Piedras Negras, Piedras Negras 26000, Mexico; rvillarreal45@hotmail.com; 7Instituto de Ciencias de la Atmósfera y Cambio Climático, Universidad Nacional Autónoma de México, Mexico City 04510, Mexico; rtorres@unam.mx; 8Sacramento Metropolitan Air Quality Management District, Sacramento, CA 95814, USA; aayala@airquality.org; 9West Virginia University, Morgantown, WV 26506, USA; 10Interdisciplinary Statistical Research Unit, Indian Statistical Institute, Kolkata 700108, India; psmukherjee.statistics@gmail.com

**Keywords:** air pollution, children, Alzheimer, APOE4, early biomarkers, amyloid beta, cognition, hyperphosphorylated tau, Metropolitan Mexico City, nanoparticles, neuroprotection, suicide, ultra-fine particulate matter, PM_2.5_

## Abstract

This Review emphasizes the impact of APOE4—the most significant genetic risk factor for Alzheimer’s disease (AD)—on peripheral and neural effects starting in childhood. We discuss major mechanistic players associated with the APOE alleles’ effects in humans to understand their impact from conception through all life stages and the importance of detrimental, synergistic environmental exposures. APOE4 influences AD pathogenesis, and exposure to fine particulate matter (PM_2.5_), manufactured nanoparticles (NPs), and ultrafine particles (UFPs) associated with combustion and friction processes appear to be major contributors to cerebrovascular dysfunction, neuroinflammation, and oxidative stress. In the context of outdoor and indoor PM pollution burden—as well as Fe, Ti, and Al alloys; Hg, Cu, Ca, Sn, and Si UFPs/NPs—in placenta and fetal brain tissues, urban APOE3 and APOE4 carriers are developing AD biological disease hallmarks (hyperphosphorylated-tau (P-tau) and amyloid beta 42 plaques (Aβ_42_)). Strikingly, for Metropolitan Mexico City (MMC) young residents ≤ 40 y, APOE4 carriers have 4.92 times higher suicide odds and 23.6 times higher odds of reaching Braak NFT V stage versus APOE4 non-carriers. The National Institute on Aging and Alzheimer’s Association (NIA-AA) framework could serve to test the hypothesis that UFPs and NPs are key players for oxidative stress, neuroinflammation, protein aggregation and misfolding, faulty complex protein quality control, and early damage to cell membranes and organelles of neural and vascular cells. Noninvasive biomarkers indicative of the P-tau and Aβ_42_ abnormal protein deposits are needed across the disease continuum starting in childhood. Among the 21.8 million MMC residents, we have potentially 4 million APOE4 carriers at accelerated AD progression. These APOE4 individuals are prime candidates for early neuroprotective interventional trials. APOE4 is key in the development of AD evolving from childhood in highly polluted urban centers dominated by anthropogenic and industrial sources of pollution. APOE4 subjects are at higher early risk of AD development, and neuroprotection ought to be implemented. Effective reductions of PM_2.5_, UFP, and NP emissions from all sources are urgently needed. Alzheimer’s Disease prevention ought to be at the core of the public health response and physicians-scientist minority research be supported.

## 1. Introduction

A growing body of evidence indicates the strong association between the Apolipoprotein (APOE4) allele, the increased risk of Alzheimer’s disease (AD), the associations with cognitive and brain volumetric changes from childhood and the increased risk for other neurodegenerative diseases besides AD [1,2,3,4,5,6,7,8,9,10,11,12,13,14,15,16,17,18]. It has been almost five decades since Shore and Shore’s [19] landmark paper on the heterogeneity of human plasma very-low-density lipoproteins and the current APOE allele differential impacting global populations [20,21,22,23]. APOE4-carrying populations were protected through evolutionary pressure against infectious diseases, including malaria, thus explaining the prevalence of allele differences across races and ethnicity [23]. Corvo et al. [20] discussed a very critical aspect of APOE allele distribution; APOE3 is the most frequent allele regardless of race and ethnicity, associated with populations with long-established agricultural economies, i.e., the Mediterranean basin, while APOE4—the ancestral allele—is higher in pygmies, Malaysia aborigines, and Native Americans with economy’s characteristic of foraging and/or food scarcity. Among Latinos in the USA [24], Caribbean Latinos from the Dominican Republic and Puerto Rico have the highest APOE4 frequency, both in normal cognitive subjects 23.2% and AD patients 32.4%, while Mexican Hispanic AD cases versus controls recorded: 21.4% and 12.5%, respectively [25]. The world APOE4 frequency varies 9–23% (i.e., Asian 9%, Hispanic 12%, white 14%, African descent 19%, other/mixed race 23%) and dramatically increases in AD patients (Hispanic 24%, Asian 28%, African descent 35%, white 38%, other/mixed race 45%) [21].

This Review emphasizes the impact of APOE4 peripheral and neural consequences starting in childhood and the environmental factors appearing to synergistically damage neural and extra neural key systems and impact the development and progression of AD [26,27,28,29]. We discuss the APOE4 effects that influence AD pathogenesis and how exposures to fine particulate matter (PM_2.5_, particles ≤ 2.5 μm), ultrafine particles (UFP ≤ 100 nm) associated with fossil fuel combustion and friction processes, and industrial manufactured nanoparticles (NPs ≤ 100 nm) are major contributors to an ample variety of neural effects, including cerebrovascular dysfunction, neuroinflammation, oxidative stress, and extensive organelle pathology and dysfunction [30,31,32,33,34,35,36,37,38,39,40,41]. We highlight hepatocytes as one of the largest contributors to the peripheral APOE pool, making the liver an organ with significant importance at times when we are facing serious liver metabolic pathologies in children and young adults [42,43,44,45].

Understanding the extensive APOE-associated detrimental neural and non-neural effects right from conception through all life stages and the importance of exposure to environmental aggressors allows the understanding of why an 11-year-old has extensive hyperphosphorylated tau in the brainstem and cortical locations, why we have diffuse beta-amyloid plaques in infants, and why APOE4 children and young adult carriers are committing suicide and accelerating their neurodegenerative processes in highly polluted environments. The challenges we face dealing with pediatric and young adults of highly exposed PM populations, particularly APOE4 carriers, include i. early identification of high-risk young AD subjects, including children; ii. making an early AD diagnosis using non-invasive biomarkers indicative of the hyperphosphorylated-tau (P-tau) and the beta-amyloid 42 (Aβ_42_) progressive neuropathological changes; iii. defining and staging the disease from pediatric ages across the heterogeneity of the AD spectrum; and iv. protecting millions of exposed people across the world. Alzheimer’s disease associated with air pollution ought to be preventable.

## 2. Peripheral APOE4 Effects

APOE impacts extra neural peripheral tissues, and it relates to their production in hepatocytes, adipose cells, kidneys, and macrophages, all contributing to the peripheral APOE pool and effects [2,19,23,35,42,43,44,45,46,47,48,49,50,51,52,53]. Martínez-Martínez and colleagues’ work is a highly recommended key reference in this section [43]. APOE polymorphic alleles directly influence plasma and brain cholesterol concentrations [43,50], and the liver is key for Apolipoprotein E (ApoE)—the cholesterol carrier crucial to lipid transport and injury repair in the brain reuptake—its release in the circulation, and impaired recycling of ApoE interfering with intracellular cholesterol transport. This interference contributes to the pathophysiological lipoprotein profile observed in APOE4 carriers and impacts the pathological accumulation of triglycerides and various lipids in hepatocytes [44,52,53]. The importance of peripheral APOE in brain AD pathology and cognition was superbly demonstrated by Liu and co-workers [35] in conditional mouse models expressing human APOE 3 or 4 in the liver, without brain APOE and abnormal synaptic brain plasticity and cognition resulting from abnormal cerebrovascular function. This work showed a very important piece of the brain APOE4 puzzle: liver expressed APOE4 exacerbates amyloid pathology in mice models [35,54]. The issue is critical for humans because, in young urbanites exposed to PM pollution, including metal, metalloid, organic, and inorganic toxic-carrying NPs [15,16,26,27,28,29,38,40], the NPs reach every tissue in the body, including the brain, blood–brain barrier (BBB), and liver. Thus, what we see as separate entities in experimental mice [35], is already in place in humans. Liver ApoE in urbanites highly exposed to NPs could readily worsen the NP BBB and brain damage, especially because the hepatocytes, Kupffer cells (KC), and sinusoidal endothelial cells are affected by NP-induced oxidative stress, sinusoidal dilatation, Kupffer cell hyperplasia, and liver inflammation [55,56,57]. Arsiwala and collaborators [58] showed increased liver monocytes and KC becoming apoptotic upon the IV injection of iron NP preparations to treat iron deficiency. The presence of nanoparticles in both hepatocytes and KC in highly exposed urbanites (personal observation Angélica González-Maciel, Rafael Reynoso-Robles, Lilian Calderón-Garcidueñas) and the acute effects on KC upon IV Fe treatment makes the issue very relevant to liver NP oxidative stress and inflammation [55,56,57,58].

Extraordinarily relevant to the associations between cardiovascular morbidity and Alzheimer’s disease is the report of Habenicht and collaborators [59] associating a C1q-ApoE complex in intimal arterial lesions and human amyloid plaques. As described by the authors, the association of C1q, an initiating and controlling protein of the classical complement cascade, is key in acute and chronic inflammatory responses and is secreted by myeloid cells (including Kupffer cells). To initiate the CCC cascade, C1q must be activated by molecules as varied as oxidized lipids, amyloid fibrils, and immune complexes, and ApoE mute towards inactive C1q binds at high affinity to its activated form. Habenicht and collaborator’s C1q–ApoE complex in intimal locations illustrates the APOE peripheral and detrimental brain effects [59].

The work of Yin and collaborators [60] showing the total hepatocyte inhibition of mitochondrial and ATP-linked oxygen consumption rate—indicative of mitochondrial dysfunction—upon exposure to diesel NPs in APOE knockout mice illustrates a key diesel effect: mitochondrial dysfunction in both the liver and brain. Yin and co-workers’ paper is of practical importance—diesel NPs are common pollutants in urban areas and occupational environments, and NPs < 100 nm readily found in the exhaust are precisely the particle size capable of crossing all biological barriers and using cation channels such as transient receptor potential (TRP) proteins involved in the regulation of intracellular biochemical signaling processes and cellular electrical excitability [27,28,29,61,62,63,64].

The phenotypic spectrum of non-alcoholic fatty liver disease (NAFLD) and ApoE was described by Meroni et al. [65] in a 40y old female with early NAFLD and severe hypertriglyceridemia. Liver biopsy showed extensive mitochondrial architecture abnormalities; thus, liver mitochondrial abnormalities are associated with genes involved in lipid metabolism and missense mutations involved in mitochondrial dysfunction. Genetic factors, such as APOE and others [65], are synergistic with metabolic-associated fatty liver disease, medications [43,46,51,55,56], and environmentally sustained and seemingly harmless nanoparticles, i.e., the case of silica, magnificently illustrated in the work of Abulikemu et al. [45]. The respiratory exposure of silica NPs induces hepatotoxicity; the authors [45] focused on the role of SiNPs in the pathogenesis and progression of NAFLD and demonstrated a significant aggravation of hepatic steatosis, inflammation, and collagen deposition in ApoE^−/−^ mice, along with high concentrations of ALT, AST, and LDH levels associated with liver damage [45]. Moreover, common NP path mechanisms are seen in liver, heart, and brain NP damage, including endoplasmic reticulum (ER) stress [66,67,68,69,70,71].

The complexity of APOE4 peripheral effects certainly also impacts body mass index (BMI), cardiovascular disease (CVD) risk, and thus, later, brain effects [72,73]. Ozen et al. showed higher fasting blood lipids and higher CVD risk in APOE4 carriers, with body composition and diet playing an important role [72]. There is no doubt that CVD risk goes hand in hand with brain aging; thus, the relationship with APOE4 becomes crucial when evaluating brain effects, as in Subramaniapillai et al. [74]. The authors evaluated 21,308 UK Biobank volunteers and established the association between white matter brain age gap (BAG) and BMI, waist-to-hip ratio (WHR), body fat percentage (BF%), and APOE4 status. Interestingly, older females [66–81 y] with greater BF% had lower BAG, while earlier menopause transition was associated with higher BAG.

APOE peripheral effects have direct and/or indirect detrimental impacts on the brain.

## 3. APOE4 Brain Effects

APOE4 is a complex risk factor for AD and other neurodegenerative diseases, with multiple, heterogenous, and overlapping action mechanisms and molecular interactions [3,5,6,10,14,30,35,46,75,76,77,78,79,80]. Stuchell-Brereton et al. [75] have shown that APOE4 is far more disordered and extended than previously described and retains conformational heterogeneity after binding lipids, which helps to understand the differences between the ε4 allele and protective variants of the protein. While there is not much controversy in stating APOE4 increases the risk of AD by driving earlier and more abundant amyloid pathology in the brains of APOE4 carriers [30], the work of Lazar et al. [76] in mice expressing the human APOE4 vs. APOE3 isoform shows APOE4 carriers had altered cholesterol turnover, ratio imbalances of specific classes of phospholipids, low phosphatidylethanolamines bearing polyunsaturated fatty acids, and an elevation in monounsaturated fatty acids. More importantly, these changes in lipid homeostasis were related to increased production of Aβ peptides and higher levels of tau and phosphorylated tau in primary neuronal cultures [76]. Tcw et al. [77] investigated the effects of APOE4 on neural cells and isogenic human-induced pluripotent stem cells, as well as in human autopsies and in APOE-targeted replacement mice. APOE4-driven metabolic dysregulation of astrocytes and microglia are major findings in their work. The significant increases in cholesterol synthesis in APOE4 astrocytes already having lysosomal cholesterol sequestration illustrates the ε4 impact associated with matrisome dysregulation. Astrocyte pathology is key in APOE4 carriers and certainly contributes to the complexity of the abnormal relationship with neurons and the neurovascular unit [78,79].

The issue of body mass has been central to APOE; indeed, once female APOE4 carriers are in the Alzheimer Continuum, a lower body mass is significantly associated with AD progression [80]. Ando and collaborators studied 1631 patients with mild cognitive impairment (MCI) or early to moderate stages of AD and described lower weight and a lower percentage of body fat associated with APOE4 in women, but not in men [80]. Remarkably, in MCI, the impact of APOE4 on women’s body composition, specifically lower fat mass, was significant, independent of cognitive decline—an observation of great importance in the day-to-day clinical assessment of patients [80]. The controversial issue of estrogen replacement in menopause and APOE4 effects must be briefly mentioned here. One major issue is the effect of hormone replacement treatment (HRT), amply discussed by Saleh et al., based on the results of the European Prevention of Alzheimer’s Disease (EPAD) cohort [81]. The authors used the baseline data from the EPAD cohort (women = 1178) and showed APOE4 HRT users had better-delayed memory index scores versus APOE4 non-users and non-APOE4 carriers, with 6–10% larger left entorhinal and bilateral amygdala volumes. Their conclusion [81] that “HRT introduction is associated with improved delayed memory and larger entorhinal and amygdala volumes in APOE4 carriers only, represents an effective targeted strategy to mitigate the higher life-time risk of AD in this large at-risk population subgroup”. We strongly support their conclusions, and their study obligates everyone interested in protecting women at risk by offering HRT when the time is right, along with protection strategies to ameliorate the APOE4 effect throughout life.

Of course, tau aggregation and hyperphosphorylated-tau play a key role in cognitive effects in the AD Continuum [82,83,84,85,86,87,88]. Critical to this Review is the Arnsten et al. [86] hypothesis that tau pathology within select projection neurons with susceptible microenvironments can initiate sporadic AD (sAD) and the equally remarkable observation by Braak and Del Tredeci [85] supporting the idea that axons of cortico-cortical top-down neurons in neocortical fields involved in AD are key for carrying and spreading abnormal tau trans synaptically into the distal dendritic segments of nerve cells following directly into the neuronal chain. The important documentation of tau seeding frequently preceding NFT pathology, e.g., in the substantia nigra pars compacta, and Bergmann glia, and the involvement of the intermediate reticular zone, dorsal raphe nucleus, and olfactory bulb [87] are of great relevance to our findings in Metropolitan Mexico City children and young adults [26,27,39,42,89,90].

Neuroinflammation plays an important role in AD and variants in APOE and the triggering receptor expressed on myeloid cells 2 (TREM2)—a microglial receptor–adaptor complex, expressed on myeloid cells 2 and which three domains are key for the binding of pathogen-associated pattern molecules PAMPS (the release of protective molecules for microglial survival and the propagation of proinflammatory signals) are linked to APOE [91,92,93]. Gratuze et al. [92] explored whether TREM2 KO (T2KO) would block neurodegeneration in P301S tau mice expressing ApoE4 (TE4) in a matter similar to microglial depletion. Instead of the expected decreased neurodegeneration and tau pathology in TE4-T2KO, the researchers found that TREM2 deficiency aggravates neurodegeneration, causes TREM2-independent microgliosis, and facilitates tau-mediated neurodegeneration in the presence of APOE4. Heneka [93], in his Comments on Gratuze et al. work [92], makes a practical key point: APOE effects have much to do with its interactions with other genetic factors, in this case, TREM2. Furthermore, in the work of Vontell et al. [94], the inflammasome sensor proteins, NOD-like receptor proteins (NLRP) 1 and 3, and caspase-1 are crucial in the early pathological stages of AD; thus, studying only targeted genetic risk factors for sAD indeed does not give us the full picture.

Zhou et al. [95] explored the leukocyte immunoglobulin-like receptor B3 (LilrB3) and its immunomodulatory effects in APOE4 versus ε2 and showed that only APOE4, specifically, interacts with LilrB3 and activates human microglia cells (HMC3) into a pro-inflammatory state in a LilrB3-dependent manner.

Inflammatory systemic responses, of course, could be very different from cerebral responses to APOE4, and Civeira-Marin et al. [96] demonstrated that APOE inflammatory modulation is independent of their lipid metabolism impact, and, in fact, their results were unexpected. On the premise that APOE and CVD negative or positive impacts are partially mediated by LDL-cholesterol, data from the Aragon Workers Health Study (n = 4159) and the Lipid Unit at the Hospital Universitario Miguel Servet (n = 3705) participated in the investigation of the relationship between C-reactive protein (CRP) levels and APOE genotype. The surprising result was that APOE4 carriers had significantly lower levels of CRP than APOE3 carriers.

The impact of APOE4 on tau early deposition in preclinical AD has been shown in cognitively unimpaired 71.2 ± 4.6 y olds [97]. APOE2 and APOE4 were associated with lower and higher amyloid positivity rates, respectively, while APOE2 was associated with reduced regional tau in all regions versus APOE4 with greater regional tau. Interestingly, the direct effects for ε4 were only significant in the medial temporal lobe. The authors [97] concluded that APOE influences early regional tau PET burden, a very important finding in view of the fact we are detecting P-tau in children and young adults in infratentorial and cortical regions in the setting of severe air pollution with extensive cognitive, olfactory, and brain structural effects [15,16,26,36,37,40,41].

Myelination changes associated with APOE impacting oligodendrocytes are part of the detrimental brain picture [98,99]. Blanchard et al. [98] compared single-cell transcriptomics profiling of human APOE4 carriers vs. non-carriers. APOE4 brain samples had altered signaling pathways associated with cholesterol homeostasis and transport. Confirming these findings, induced pluripotent stem-cell-derived cells and targeted-replacement mice showed that cholesterol is aberrantly deposited in oligodendrocytes-myelinating cells and coincides with reduced myelination.

Mock et al. [99] showed, on the other hand, that APOE4 impairs myelination in the aging brain by interrupting the delivery of astrocyte-derived lipids to the oligodendrocytes (OL). The authors propose that high myelin turnover and OL exhaustion in APOE4 carriers was a plausible explanation for the APOE-dependent myelin phenotypes of the AD brain [99]; however, we have shown white matter changes in APOE4 and, to a lesser degree, in ε3 pediatric and young adults. Therefore, in the setting of severe air pollution, myelin and axonal changes relate to direct oligodendroglia and axonal damage and indirectly to severe BBB damage [41,100,101].

### 3.1. Mitochondria, Energy Generation, the Ubiquitin Proteasome System, and Proteotoxicity

Mitochondrial dysfunction is key for neurodegenerative diseases [102,103,104,105,106,107,108,109,110,111,112,113,114,115,116,117,118,119,120,121,122,123,124,125,126,127,128,129,130]. Mitochondria are at the core of cellular energy production and interruption of the crosstalk between cell signaling and mitochondrial functions and biogenesis, and result in mitochondrial dysfunction, proteotoxic stress, accumulation of defective mitochondria, and the production of reactive oxygen species due to defective mitophagy and major alterations in the ubiquitin–proteasome system [102,103,104,105,106,107,108,109,110,111,112,113,114,115,116,117,118,119,120]. The integrity of the double mitochondrial membranes with distinct functions, including communication with the intracellular microenvironment and the organelle contact sites, is the gate for the entry of precursor proteins produced on cytosolic ribosomes, as reviewed by den Brave et al. [102]. The ubiquitin–proteasome system is a crucial player in protein surveillance on the mitochondrial surface [102]. Proteotoxic stress, defined in Karbowski et al. [103] as a reduction in bioenergetic activity induced by the accumulation of aberrant proteins in the mitochondria, is a mechanism at play in several neurodegenerative diseases. Mitochondrial unfolded protein responses (mtUPR), the ubiquitin (Ub)-dependent degradation of aberrant mitochondrial proteins, and mitochondria-specific autophagy (mitophagy) all respond to proteotoxic stress and eliminate defective proteins or dysfunctional mitochondria. Failure to accomplish perfect mitochondrial quality control mechanisms has fatal brain consequences [104,105]. Abnormalities in the ubiquitin–proteasome system and mitophagy result in defects of brain energy translocation and accumulation of the byproducts of oxidative stress [104,105,106,114,115,116,117,118,119,120,121].

Mitochondrial dysfunction is also potentially impacted by the differences in mitochondrial DNA methylation associated with age, sex, and anatomical brain regions [107] and the capacity of glial cells, including oligodendroglia, to distribute mitochondria to subcellular locations [108]. Since myelination is an energy-consuming event, as described by Gil et al. [108], oligodendrocyte development and mitochondrial-mediated mechanisms to oligodendrocyte bioenergetics and development, as well as mitochondrial development in utero are also relevant to neurodegeneration [109,110,111,112]. Inflammatory environments negatively impact mitochondria, as shown by Liao et al. [113], using conditioned medium from LPS-activated macrophages in mice hypothalamic neurons resulting in elevated intracellular ROS/RNS levels and reduced antioxidant enzymes. Interestingly, LPS exposures not only affected mitochondrial respiration and functions, but also resulted in the elongation of mitochondria after a 24 h treatment. The authors concluded that LPS chronic exposure significantly increased oxidative stress, decreased mitochondrial respiration, and altered mitochondrial dynamics [113]—the issue is of utmost importance in urbanites exposed to LPS-associated fine particulate matter [123].

A number of authors coincide in the role of APOE4, defective mitochondria, and proper scavenging of free radicals through a number of paths, including oxidative phosphorylation, increased contact and apposition of mitochondrial-endoplasmic reticulum ER membranes, damage to mitochondria-associated endoplasmic reticulum (ER) membranes (MAM), autophagy impairment, mitochondrial biogenesis and dynamics, and mitophagy and mitochondrial DNA changes [123,124,125,126,127,128,129,130]. APOE4 carriers are unable to properly scavenge 4-hydroxynonenal (HNE)—a highly reactive and neurotoxic molecule—due to the lack of Cys residues; the result is the oxidation of neuronal proteins and neuronal death [130]. Butterfield and Masson discussed that this excess of HNE in APOE4 carriers is, in fact, a major factor for neuronal oxidation from early neurodegenerative stages [130].

Impairments of neuronal insulin associated with the trapping of the insulin receptor in endosomes [131] link APOE4, diabetes mellitus, and AD-associated amyloid pathology. APOE4 alters neuronal insulin signaling in human APOE-targeted replacement (TR) mice and APOE4-TR mice fed a high-fat diet, accelerating these effects in middle age [131]. APOE4 interacts with the insulin receptor and impairs its trafficking by trapping it in endosomes, leading to impaired insulin signaling and insulin-stimulated mitochondrial respiration and glycolysis [131]. The disruption in the fatty acid regulation alters both neurons and astrocytes and decreases FA astrocytic transport [132]. APOE4 status lowers FA oxidation and leads to lipid accumulation in both astrocytes and the hippocampus, accelerating lipid dysregulation and energy deficits and increasing AD risk [132], a situation that worsens with age [133].

### 3.2. Ethnicity and APOE4

Ethnicity matters for APOE4 effects [134]. In 449 subjects from the UC Davis Diversity Cohort who had APOE genotyping and ≥2 cognitive assessments, Chan et al. [134] described an ϵ4 prevalence in black (46%) and white (46%) versus Latino participants (24%). APOE4 was associated with poorer baseline episodic memory only in white participants (*p* = 0.001), but all racial/ethnic groups exhibited a moderately strong association with episodic memory change. Hispanics represent varied ethnic and cultural backgrounds in the USA and, as such, respond differently, as shown by Llibre-Guerra et al. [135]. The researchers explored 13,516 subjects (5198 men, 8318 women) with an average age of 74.8 years with data regarding APOE genotype, dementia prevalence, and memory performance (immediate and delayed recall scores) in Caribbean Hispanics (CH), African Americans (AA), Hispanic Americans (HA), and non-Hispanic White Americans (NHW). Interestingly, and different from Chan et al. [134] in California, the APOE4 prevalence was similar across ethnicity (21.8–25.4%), except for African Americans (33.6%; *p* ≤ 0.001) [135]. APOE ε4 carriers showed higher dementia prevalence across all groups [135]. It is clear that a healthy lifestyle is a protector of cognitive decline [136]. Dhana and coworkers [136] examined 3874 African Americans and European Americans genotyped for APOE from the Chicago Health and Aging Project (1993–2012; n = 3874) and explored a healthy lifestyle using: non-smoking, exercising, being cognitively active, having a high-quality diet, and limiting alcohol use. Although APOE4 was associated with faster cognitive decline, African Americans who had 4/5 healthy factors exhibited a slower cognitive decline (*p* = 0.023) versus European Americans (*p* = 0.044). Mexican Americans have a frequency of ε4 (17.5%) and ε2 (6.6%) alleles versus Non-Hispanic Whites NHWs (ε4 27.9 and ε2 15.9) [137], and the ε4 allele was associated with poorer cognition across multiple domains among NHWs; however, Mexican Americans with fewer years of education showed poorer memory performance. This information is critical for US physicians since Mexican Americans have less education and less access to good health care and jobs where they are highly exposed to neurotoxins.

## 4. APOE4 in Children and Young Adults: Air Pollution Plays a Major Role in the Cognitive, Behavioral, and Brain Structural Responses in Pediatric APOE4 Carriers

The association between APOE4 and increased cholesterol, triglycerides, and lipoprotein levels in children has been described for more than two decades, and metabolic syndrome has been particularly prominent among females with 3/4 genotypes [138,139,140,141,142]. APOE4 girls also have hypertension and low HDL [140]. Mexico City APOE 3/4 girls, aged 12.3 ± 5.4 years, exhibited decrements in attention and short-term memory and below-average scores in Verbal, Performance, and Full-Scale IQ. Strikingly, APOE4 heterozygous girls with >75% to <94% BMI percentiles were at the highest risk of severe cognitive deficits (1.5–2 SD from average IQ) [141,142].

APOE4 differential responses start in infants born preterm, and Dzietko et al. [143] and Humberg et al. [144] described intraventricular hemorrhage in APOE4 carriers and an increased risk of motor deficits after intracerebral hemorrhage. Lien et al. [145] studied children from the Norway Cerebral Palsy Register and described an APOEε4 adverse effect on the developing brain after injury.

APOE carrier status clearly impacts neurobehavioral performance, cognition deficits, and sleep in pediatric populations across the world [146,147,148,149,150,151,152,153,154,155]. Remarkably, exposures to traffic air pollution during pregnancy play a key role in neurodevelopmental outcomes [17]. Alemany et al. [17] published a paper focusing on traffic-related pollution, behavior problems, ADHD symptom scores, cognitive performance trajectories, and the link with 1667 children’s APOE4 status. Their findings are very noteworthy, especially because they used ambient concentrations of pollution measured in Barcelona and surrounding towns. Specifically, researchers examined concentrations of outdoor polycyclic aromatic hydrocarbons (PAHs), elemental carbon (EC), and nitrogen dioxide (NO_2_) measured at school sites (two 1-week campaigns conducted 6 months apart), and pollution levels were found to be below the USEPA annual standards. However, their PM_2.5_ of 16.7 ± 3.16 μg/m^3^ was above the USEPA annual standard (12 μg/m^3^) [156]. Behavior problems were significantly higher in APOE4 carriers, and there was a negative association with PAHs and NO_2_; moreover, smaller caudates were significant for APOE4 children. Caudate atrophy is indeed a strong MRI finding across young Mexico City residents, with much higher PM_2.5_ concentrations than Barcelona [41].

The higher the exposure to pollutants, especially PM_2.5_, the stronger the detrimental impact on children [147]. Holm et al. described the longitudinal associations between prenatal PM_2.5_ exposure and measures of IQ in 568 children, with an average age of 10.5 y. These children were the sons and daughters of Mexican agricultural workers in Salinas, CA, with low socioeconomic status (SES) [147]. The authors’ results showed that lower IQ was associated with relatively small increases in PM_2.5_ exposures (3 μg/m^3^) in mid to late pregnancy, and boys were more affected than girls. Factors such as lack of stimulation in an environment that does not favor academic achievement and households in the poverty level are key factors in determining cognition, aside from air pollution. ADHD, autism, and DNA methylation changes also have strong links with pregnancy exposure across the world [153,154,155] and involve combustion sources, including indoor wood burning and road traffic.

### 4.1. Brain MRI in APOE4 Children and Young Adults

The association between high concentrations of air pollutants, cognition deficits, and brain MRI alterations was established in 2008 when matched SES, mothers’ education level, age, and gender in 55 children from Mexico City versus 18 from a low polluted city at slighter higher altitude over sea level showed prefrontal white matter hyperintense lesions and similar lesions were observed in young Mexico City dogs (57%) [36]. Mexico City children (average age 10.3 ± 2.1 y) exhibited deficits in fluid and crystallized cognition tasks [36]. Vascular subcortical arteriolar pathology associated with neuroinflammation, enlarged Virchow-Robin spaces, periarteriolar gliosis, and intracellular ultrafine particle (UFP) deposition were the major neuropathology findings in dogs and were confirmed later in children and young adults [26,27,28,37,38,39,40]. In 2011, a one-year follow-up study of global and regional brain volumes of Mexico City versus low pollution control children showed white matter volumetric differences in right parietal and bitemporal areas in keeping with progressive deficits on the WISC-R Vocabulary and Digit Span subtests [157]. Interestingly, at that time, the presence of white matter hyperintense lesions only partially correlated with progressive cognitive deficits [157]. We know now that the vascular white matter lesions are dominated by the presence of NPs in endothelial cells, causing tight junction abnormalities, and white matter lesions are extensive and severe from very early infancy [38,89,100,101]. Indeed, vascular brain damage in utero starts very early, and by age 12–15 postconceptional weeks, we can document the transfer of NPs from erythroblasts to the brain vascular endothelium [29]. Alexopoulos et al. [158] published a brain MRI work showing that healthy APOE4 carriers aged 24.4 ± 3.8 y with 16.9 years of formal education living in Erlangen-Nuremberg had statistically significantly smaller hippocampal volumes than APOE2 carriers. No memory differences were detected between the two groups. Interestingly, Erlangen-Nuremberg [159] has PM_2.5_ above the annual USEPA standard [156]. In 2012, O’Dwyer et al. published the results of their Frankfurt study in a healthy cohort of 44 people 26.8 ± 5.2 y (22 APOE4 carriers), and measured volumes of accumbens, amygdala, caudate, putamen, pallidum, thalamus, hippocampus, and brainstem [160]. A significant reduction in volume was documented in the right hippocampus of APOE4 carriers vs. non-carriers. As in Alexopoulos’ paper [158], no differences in memory were detected [160]. The authors suggested that early hippocampal atrophy may be occurring in APOE4 carriers, but actual memory performance is not yet undermined.

Remarkably, in 2014, Dean et al. [161] studied 162 healthy, 2–25-month infant APOE4 carriers versus non-carriers from mothers with limited formal education [162] and quantified white matter myelin water fraction (MWF) and gray matter volume (GMV) [161]. Their findings were very interesting; ε4 carriers had lower MWF and GMV measurements in precuneus, posterior/middle cingulate, lateral temporal, and medial occipitotemporal regions versus non-carriers. Larger MWF and GMV measurements were also recorded in frontal regions in 2–6-month-old APOE4 babies. These children were exposed to annual PM_2.5_ concentrations of 9.7 µg/m^3^ [163]. The authors suggested that the MWF and GMV changes in APOE4 infants are the earliest brain changes associated with the genetic predisposition to AD [161].

Khan et al. [164], in a study of 1400 adolescents ages 14.45 ± 0.41 years from the European multi-center neuroimaging-genetics IMAGEN project, found no differences in hippocampal volume or asymmetry differences between carriers and non-carriers of APOE4 or ε2 alleles. Meanwhile, Chang et al. [165] studied APOE genotypes in 1187 children aged 12.1 ± 5.0 y from the Pediatric Imaging Neurocognition and Genetics Study, aiming to define if there were differences in gray matter maturation according to APOE status. In this cohort, 3/3 was the most common genotype, followed by 3/4 (21.8%), 2/3 (11.9%), 2/4 (2.6%), and 4/4 (1.75%). The results showed small hippocampi in 2/4 children, the lowest hippocampal fractional anisotropy in younger 4/4, the largest medial orbitofrontal cortical areas in 3/4 children, and thinning of the entorhinal cortex in 4/4 children [165]. The authors concluded, “defining APOE ε polymorphisms in young children may provide the earliest indicators for individuals who might benefit from early interventions or preventive measures for future brain injuries and dementia” [165]; we fully agree with them.

Remer et al. [166] studied 223 children aged 2–68 months, including 74 ε4 carriers and 149 non-carriers, matched for age, gestational duration, birth weight, sex ratio, maternal age, education, and SES, followed longitudinal MRI white matter myelin and cognitive-behavioral changes in APOE4 vs. non-carriers. APOE ε4 carriers had significant differences in white matter myelin development involving the uncinate fasciculus, temporal and occipital lobes, and internal capsule. APOE4 carriers had a greater rate in early learning composite—an IQ surrogate—but lower non-verbal development of fine motor and visual skills across the age range (up to 68 months) [166].

The cognitive impact of APOE4 in young carriers follows the antagonistic pleiotropy hypothesis [167] at younger ages, as shown by Zink et al. [168] in 190 healthy German 23.81 ± 3.2 y subjects. APOE ε4 carriers were able to better adapt to different degrees of cognitive control requirements, with superior performance in case of high control demands associated with modulations of the N450 component rooted in superior frontal gyrus activation differences. The authors concluded that young ε4 carriers are more efficient at allocating cognitive control resources based on the actual task requirements. APOE4 subjects “experience less conflict, exert less effort, and recruit fewer additional prefrontal areas when task set complexity increases” [168].

### 4.2. Magnetic Resonance Spectroscopy (MRS) and APOE4

Neurometabolic brain changes are described in brain spectroscopy using magnetic resonance (MRS), and both 31 phosphorus magnetic resonance spectroscopy (^31^P-MRS) and proton MR spectroscopy (^1^H-MRS) are common techniques applied to study the APOE4 impact on neurometabolites and their abnormalities in Alzheimer’s patients, including sporadic and autosomal dominant AD [169,170,171,172,173,174,175,176].

Subjects with autosomal dominant AD have lower levels of N-acetyl-aspartate+N-acetyl-aspartyl-glutamate (NAA) and glutamate/glutamine (Glx) in the left pregenual anterior cingulate cortex and lower levels of NAA and higher levels of choline (Cho) and myo-inositol (mI) in the precuneus compared to controls [169]. Interestingly, Joe et al. [169] described high mI in the precuneus and left pregenual anterior cingulate cortex as the subjects were getting closer to the expected age of dementia onset. Remarkably, the MRS was useful in showing that the metabolic changes during the preclinical period were very similar to those seen in late-onset AD [169].

Jett et al. [170] used ^31^P-MRS to examine the effects of APOE4 status on high-energy brain phosphates (adenosine triphosphate (ATP), phosphocreatine (PCr), inorganic phosphate (Pi)) and membrane phospholipids (phosphomonoesters (PME), phosphodiesters (PDE)). Two-hundred and nine cognitively intact subjects aged 40–65 y, 46% APOE4, were included in the study. Women were more affected than males, and PCr/ATP and PCr/Pi levels were lower, while the APOE4 cohort showed lower PCr/ATP and PCr/Pi in frontal regions as compared to non-carriers. The authors [170] emphasized the higher risk of midlife women versus men and suggested mitochondrial high-energy phosphates support the identification of risk groups, in agreement with Parasoglou et al. [171].

Neurometabolite alteration precedes the detection of amyloid pathology [174], and in APOE4 carriers, the levels of creatine can be significantly lower vs. 3/3 carriers, suggesting increased brain metabolic demands [175]. In the work of Laakso et al. [175], creatine correlated significantly with age and Mini-Mental State Examination test performance only in APOE4 carriers.

Female sex is a powerful risk factor for AD and thus moderates the relationship between hippocampal GABA+ and episodic memory, resulting in women having lower GABA+ concentration and worse memory performance [172]. These findings in a small sample size had no association with APOE4 [172].

There is no question that there is an urgent need to identify early AD cases, and Chen et al. [176] aimed to define the posterior cingulate ^1^H-MRS patterns in biologically defined AD (amyloid and tau +). Their study included 37 controls, 16 early AD and 15 late AD [176]. Early AD patients showed lower NAA/Cr (*p* = 0.003) versus late AD with lower NAA/Cr, and higher mI/Cr. The lower NAA/Cr was significantly associated with global amyloid and tau load.

Kara et al. [177] cognitively intact volunteers also showed an increase in posterior cingulate gyrus tau deposition associated with lower NAA/tCr and glutamate (Glu)/tCr ratios. The authors concluded that “higher tau levels in cognitively unimpaired older adults are associated with biomarkers of neural and synaptic injury even in the absence of cognitive impairment and these relationships appear to be stronger in women than in men” [177].

Abnormal neurometabolites appear early in the AD process and occur within the spectrum of Alzheimer’s Disease, including early changes in APOE4 carriers; validation between AD biomarkers and APOE4 was reviewed by Piersson et al. [178]. In their review, they found targeted brain areas, i.e., posterior cingulate cortex/precuneus, having lower levels of NAA, NAA/Cr, and NAA/mI, along with increases in mI, mI/Cr, and mI/NAA. Tau levels were associated with increased mI and reduced NAA/Cr, NAA, and glutathione levels lower in APOE4 carriers, and there was an interaction between APOE4, Aβ42, and mI/Cr [178]. The relationship between AD biomarkers and MRS findings is certainly seen in cognitively intact older adults, as shown by Hone-Blanchet et al. [179] in frontal metabolites. One hundred and forty-four normal subjects had frontal gamma-aminobutyric acid (GABA+) and mI/tCr predicted by age. Moreover, tNAA/tCr ratios were predicted by tau.

## 5. APOE, Alzheimer’s Disease, Air Pollution, and Nanoparticles: The Impact of Air Pollution on the Progression of Biological Alzheimer’s in Pediatric and Young Adult Metropolitan Mexico City Residents

Alzheimer’s Disease is a highly heterogeneous multifactorial disorder [180], defined by the National Institute on Aging and Alzheimer’s Association Research Framework [181] “by its underlying pathologic processes that can be documented by postmortem examination or in vivo by biomarkers”. Thus, the confirmation of AD diagnosis requires AD pathology biomarker evidence and specific AD clinical phenotypes [181,182]. Frisoni and colleagues [183] proposed a probabilistic model of AD with autosomal dominant APOE4-related sporadic and APOE4-unrelated sporadic AD, where, in non-dominant variants, environmental exposures and lower-risk genes play a key role. In addition, we know that overlapping pathologies, including α-synucleinopathy and vascular and TAR DNA-binding protein 43 (TDP-43) pathology are present in AD patients [182,184].

Residents in highly polluted cities such as Metropolitan Mexico City (MMC) are chronically exposed to high concentrations of a complex mixture of air pollutants dominated by fossil fuel burning, fine particulate matter (PM_2.5_), and UFP [185]. Residents have been exposed to PM_2.5_ above the annual USEPA standard (12 μg/m^3^) for decades (Figure 1A), and UFP/NPs are generated by vehicular emissions and industrial sources (Figure 1B).

Forensic autopsies in MMC children and young adults exhibit the neuropathology hallmarks of Alzheimer’s and Parkinson’s diseases and TDP-43 pathology [26,27,39,40], and metal and metalloid UFP/NPs are seen in intracellular locations in association with damaged mitochondria, endoplasmic reticulum, Golgi apparatus, heterochromatin, etc., involving the brain and heart tissues [26,27,28,71,89,100,101]. We have also documented early, progressive neurovascular unit damage and key organelle ultrastructural endothelial pathology associated with NPs [38,42,89]. Breakdown of the BBB can be documented in infants, and NPs extensively damage both endothelial cells and their tight junctions [26,27,38,42]. Interestingly, MMC organic PM_2.5_ components include lipopolysaccharides (LPS); thus, neuroinflammation and BBB remodeling-associated effects, as described by Erickson et al. [190] in LPS-treated mice, are seen in MMC residents.

AD neuropathology markers were present in 202 of 203 MMC autopsies in subjects ≤ 40 y old, including 44 children, as illustrated in Figure 3 of ref. [26]. These subjects had no extra neural light microscopy pathology.

All children in the first decade of life had P-tau pre-tangle stages, and by the second decade, we documented neurofibrillary (NFT) tangles I–V [26,191] (Figure 2). Subjects in the fourth decade were in NFT I–V stages, and we could no longer document pre-tangle stages. In contrast, Aβ progressed slowly and was kept in the early phases. Remarkably, in our autopsy studies [26], APOE4 carriers had higher NFTs Braak stages, and the highest risk for suicide was associated with lower cumulative exposures to PM_2.5_ versus APOE3 carriers (Figure 3).

We have described the remarkable AD, PD, and TDP-43 pathology overlap in young MMC residents and the fact that it is very similar to the mixed protein pathologies in elderly patients with AD, frontotemporal lobar dementia (FTLD), Lewy body disease (LBD), Parkinson’s disease (PD), amyotrophic lateral sclerosis (ALS), cerebral amyloid angiopathy (CAA), and in younger than 60 y AD patients [180,184,192,193,194,195,196]. Hyperphosphorylated-tau is the major aberrant protein in MMC young residents; P-tau in substantia nigra and lack of nuclear TDP-43 in cortical motor neurons, motor neurons for cranial nerves III, V, and XII, and cervical motor neurons in teens and young adults were common findings [26,27,28]. Figure 4 shows the aberrant neural protein overlap in MMC 186 young residents compared [27] to elderly subjects in 375 autopsies conducted by Karanth et al. [184].

Localization of highly toxic Fe, Ti, Hg, W, Al, and Zn spherical and acicular NPs in the locus coeruleus (LC), neural and vascular mitochondria, endoplasmic reticulum, Golgi, neuromelanin, heterochromatin, and nuclear pore complexes along with early and progressive neurovascular damage is of utmost importance for MMC young residents [89] in view of Mercan and ‘s work [197]. Mercan and Heneka emphasized LC neuronal loss as one of the earliest indicators of neurodegeneration in AD. The authors commented on how LC degeneration results in decreased noradrenalin levels, increased neuroinflammation, enhanced amyloid and tau burden, cognition impairment, and decreased long-term synaptic plasticity. Alterations in the locus coeruleus–noradrenaline system are important contributors to AD progression, and indeed, severe loss of LC cells and subcellular damage associated with NPs are present in young children in MMC [27,89,197]. Remarkably, FeNPs 4 ± 1 nm and Hg NPs 8 ± 2 nm were seen predominantly in the LC and substantia nigra (SN) [27,89]. NP damage to the substantia nigrae and cerebellum is outstanding and translates to autonomic dysfunction, gait and balance alterations, and cerebellar MRI atrophy in young MMC residents [41,89,90,101,198].

Nanoparticles were also present at all placental stages, including 8–15-week placentas [29]. NPs were documented in maternal red blood cells (RBC), early syncytiotrophoblast, Hofbauer cells, and fetal endothelium (ECs). Fetal ECs displayed caveolar NP activity and widespread erythroblast-loaded NP contact. Erythroblasts are the main carriers of NPs to the developing brain. Primitive neural cells showed nuclear, organelle, and cytoplasmic Fe, Ti, and Al alloys, and Hg, Cu, Ca, Sn, and Si NPs in both singles and conglomerates. Combustion-derived NPs, as well as industrial NPs, are documented in early fetal brains [29].

## 6. CSF Alzheimer’s and TDP 43 Pathology Markers in MMC Children and Young Adults

MMC children aged 11.2 ± 5.5 y have key AD biomarkers in CSF samples, including significantly lower Aβ_42_ (*p* = 0.005) versus clean air age- and gender-matched controls [199]. Brain-derived neurotrophic factor (BDNF) concentrations are also lower in MMC children [199]. CSF cytokines and chemokines, including macrophage inhibitory factor (MIF), IL6, IL1Ra, IL-2, and Cellular prion protein PrP(C), are increased in MMC children versus controls and likely reflect the short- and long-term effects of systemic inflammation and dysregulated neural immune responses upon prolonged exposure to air pollutants [200]. CSF non-P-tau showed a strong increase with age significantly faster among MMC versus controls (*p* = 0.005) and was associated with the anterior cingulate cortex showing a significant decrease (*p* < 0.0001) in the average axonal size [201]. Significant age increases in non-P-tau support tau changes early in a population with axonal pathology and evolving AD hallmarks in the first two decades of life [201].

CSF TDP-43 exponentially increases with age (*p* < 0.0001) and reaches higher concentrations for MMC residents [202]. TDP-43 cisternal CSF levels of 572 ± 208 pg/mL forecasted TDP-43 pathology in the olfactory bulb, medulla and pons, reticular formation, and motor nuclei neurons in MMC autopsy cases [40].

Thus, the striking lower CSF Aβ_42_ concentrations fulfill one of the A+T-(N)- Alzheimer’s pathological changes at age 11.2 ± 5.5 y [181]. The low CSF Aβ_42_ in MMC children vs. controls was expected based on the neuropathology staging using Aβ_42_ [26].

## 7. Cognition Deficits in MMC APOE4 Children Are Significantly Higher than APOE3

Significant cognitive deficits involving a combination of memory, executive functions, and fluid cognition using WISC-R for children were documented in 10.7 ± 2.7 y MMC healthy children vs. controls [36]. Several studies in MMC children have confirmed the original findings and were expanded to explore APOE4 carriers [15,142,157]. Overall, MMC children performed below their normative level of cognitive development versus matched clean air controls, and remarkably, APOE4 heterozygous females showed the highest risk of severe cognitive deficits [142].

The combination of WISC-R and the University of Pennsylvania Smell Identification Test (UPSIT) administered to 50 MMC children (13.4 ± 4.8 years, 28 APOE ε3 and 22 APOE ε4) showed APOE4 children had a reduced NAA/Cr ratio in the right frontal white matter and decrements on attention, short-term memory, and below-average scores in Verbal and Full-Scale IQ (>10 points) [15]. APOE strongly modulated the group effects between WISC-R and left frontal and parietal white matter and hippocampus metabolites. Failing to ID soap in the UPSIT strongly correlated with a left hippocampus mI/Cr ratio. We concluded that APOE modulates responses to air pollution in the developing brain, and APOE4 carriers could have a higher risk of developing early AD if they reside in MMC [15]. We now know that cognition changes progress in early adulthood and healthy adults aged 21.6 ± 5.8 years, with 13.69 ± 1.28 formal education years, have an average Montreal Cognitive Assessment (MoCA) score of 23.9 ± 2.8, and 24.7% and 30.3% scored ≤ 24 and ≤22, respectively (MCI ≤ 24, dementia score ≤ 22) [202]. As people stay in MMC, the cognition worsens [203]—in a cohort of 336 clinically healthy, middle-class, Mexican volunteers aged 29.2 ± 13.3 years with 13.7 ± 2.4 years of education, the >31-year-olds had MoCA on average 20.4 ± 3.4 vs. low pollution controls 25.2 ± 2.4 (*p* < 0.0001). The number of formal years of education positively strongly impacted MoCA total scores across all participants (*p* < 0.0001) [203].

MMC young residents also have a significant risk of falling due to alterations in gait and equilibrium [90,198]. Central and peripheral auditory system dysfunction with abnormal brainstem auditory-evoked potentials starting in children 8.52 ± 3.3 y, young adults (21.08 ± 3.0 y), and middle-aged subjects 42.48 ± 8.5 y is likely related to the presence of P-tau, Aβ_42_, and α-synuclein within auditory and vestibular nuclei, along with the marked dysmorphology in the ventral cochlear nucleus and superior olivary complex [204,205,206].

Certainly, the MRI findings in young health MMC adults [41] show significant hemispheric differences in frontal and temporal lobes, caudate nucleus, and cerebellar gray and white matter in ≤30 vs. ≥31-year-olds and strong associations between MoCA total and index scores and caudate bilateral, frontotemporal, and cerebellar volumetric changes. In keeping with the children’s cognitive deficits, MoCA Language Index Scores (LIS) in young adults are significantly affected. Moreover, there are overlapping patterns of brain atrophy described for AD, PD, and frontotemporal dementia [41].

## 8. Summary

APOE4 is a key genetic risk factor for sporadic Alzheimer’s Disease (sAD) and has a significant detrimental impact on children and young adults subject to sustained PM_2.5_ and UFP/NP exposures [26,27,28,29,36,37,38,39,40,41].

We have shown the presence of P-tau and Aβ_42_ in Metropolitan Mexico City (MMC) 202 of 203 forensic autopsies in a 5-year sampling period in 11 month to 40 y old subjects with no major extra neural pathology. The AD hallmarks are already present in MMC infants, and progression is seen as the subjects grow up in a complex mixture of toxic pollutants. The cognition changes are documented in the first two decades of life, along with brain structural changes, abnormal gait and equilibrium, and BAEPs, supported by auditory nuclei dysmorphology, brainstem accumulation of aberrant neural proteins, and high concentrations of magnetic NP in the atrophic cerebellum. The overlap of P-tau, Aβ _42_, α-synuclein, and TDP-43 is documented in the 202 autopsies and the brain MRI in young, seemingly healthy volunteers showing cortical and subcortical atrophy overlapping AD, PD, and FTLD patterns, including a striking caudate and cerebellar atrophy.

These quadruple pathologies affect both APOE3 and 4 MMC residents, and noteworthily, the progression to AD accelerates in APOE4 individuals by the beginning of the third decade of life [26,101]. Overweight APOE4 girls are at high risk of metabolic syndrome, with IQ losses of ≥10 points [141,142].

A major concern in children and young adults is their subjective memory complaints and their related academic problems at school in keeping with their combination of memory, executive functions, and fluid cognition deficits documented in 10.7 ± 2.7 y MMC healthy children vs. controls [36,101,142,157] and Montreal Cognitive Assessment (MoCA) in the mild cognitive impairment (MCI) and dementia ranges in young adults [201,202].

The issue of young APOE4 males at high risk for suicide over their APOE3 counterparts [26] should be emphasized. First, males are more likely to commit suicide using weapons or other means, with high chances of death outcomes (i.e., hanging), and thus the forensic data show mostly males; their key finding was their APOE4 status, and how, at the same cumulative PM_2.5_ exposures as their APOE3 counterparts, they had strikingly accelerated P-tau stages [26]. This is not a surprising finding; AD and AD-related dementia (ADRD) patients are at a higher risk of suicidal behaviors, as shown by Alipour-Haris et al. [207] in 12,538 US hospitalizations (2016–2018) related to suicidal behaviors and in suicidal ideators with late-life depression in whom TNF-α, APOE ε4 allele presence, CRP, and high-density lipoprotein cholesterol contributed most to suicidal ideation [208]. The work of Abramova et al. [209] is extraordinarily important for megacities’ exposed populations. They genotyped 146 healthy controls and 112 dementia patients for APOE *rs429358*, *rs7412,* and genes associated with suicide—an association with dementia of *rs4918918* (*SORBS1*) and *rs10903034* (*IFNLR1*) previously associated with suicide was established, and they confirmed the association of *APOEε4* with suicide.

In sharp contrast with the management of elderly subjects, in MMC, we are dealing with a young population with progressive cognitive deficits, unable to achieve optimal performance at all levels of education. Although there is a significant cognitive advantage for those young adults with the most years of formal education, cognition deficits, and/or MRI brain atrophy are documented in MMC healthy young volunteers regardless of APOE status. This observation has a high impact in a country where a reduced number of students obtain bachelor’s degrees [210] and has the lowest tertiary attainment rate (27%) among 25–34-year-olds in the Organization for Economic Cooperation and Development (OECD) [211]. In Mexico, only 31% of individuals reported having basic ICT skills (i.e., activities such as knowing how to send an email with an attachment), which is less than the OECD average (55%); Mexican adults 18–24 y who are neither employed nor in formal education or training (NEET) reached 22% in 2021 [211].

The data we generated using MoCA in healthy young adults with college education indicates that subjects aged 21.60 ± 5.8 have a total MoCA of 23.9 ± 2.8, while those aged ≥31 y had an MoCA average of 20.4 ± 3.4 (vs. low pollution controls 25.2 ± 2.4 *p* < 0.0001). This striking difference in the MoCA scores puts ~74% of the MMC young adult middle-class population experiencing cognition deficits [203].

In the NIA-AA research framework [181], the term Alzheimer’s disease (AD) refers to pathological processes in a living person defined by biomarkers [181]; thus, we are in a very precarious situation as we are unable to use any of the invasive biomarkers in our young populations. Since the AT (N) system is designed to incorporate new biomarkers beyond those currently in place, there is an urgent need to have noninvasive biomarkers of P-tau and Aβ_42_ abnormal protein deposits across the disease continuum starting in childhood.

If we focus on AD as a continuum as per NIA-AA [181], i.e., progressive cognitive decline along the progression of biomarker measures, we have APOE3 and 4 MMC children and young adults in the Alzheimer continuum from the first decade on, with a striking significant accelerated progression in young APOE4 carriers. The risk of progressing to P-tau stage 3 or higher is significant for APOE4 subjects (*p* = 0.0089), while the risk of progressing to Aβ phase 3 or above is also higher for APOE4 subjects (*p* = 0.0035) after adjusting age and gender [26].

MMC residents have abnormal protein deposits that define biological AD in combination with α-synuclein and TDP-43 pathology, which complicates the cognitive scenario [180,184]. Figure 5 shows the differences between the staging of P-tau and Aβ_42_ in APOE3 versus APOE4 MMC ≤ 40 y old residents and the MoCA total scores [82,83,84,85].

Thus, essentially in MMC subjects, the abnormal protein deposits that define AD are present from infancy, and the AD Continuum is seen in pediatric, teen, and young adult populations. The critical periods for interventions, particularly for APOE4 carriers, are the first two decades of life. As the current research framework for AD does not incorporate air pollution parameters, there is an urgent need to include cumulative particulate matter exposures from conception and to have access to PM_2.5_ and nanoparticle data, and their composition.

APOE4 is clearly playing a key role in AD pathogenesis (Figure 5). Neuropathological and clinical, behavioral, and brain structural image changes start in pediatric ages.

The current global estimates of people in early AD stages (prodromal and preclinical) [213] are no longer valid without taking lifelong environmental air pollution and occupational exposure into account and, in particular, nanoparticle exposure—which is rarely measured and certainly not regulated in the US.

We have several proposals for the study of populations exposed to high PM_2.5_ and nanoparticle concentrations from intrauterine life:We need to longitudinally monitor pediatric populations with non-invasive AD biomarkers reflective of the neuropathological changes.We have evidence of structural brain changes in young children, and metabolic changes are readily present in APOE4 children and their parents, which offers available non-invasive (N) neurodegeneration/neuronal injury markers. We need to implement MRI and MRS longitudinal studies contrasting children in high- versus low-polluted cities, matching SES, mother’s IQ, educational level, age, gestational duration, breastfeeding history, birth weight, sex, maternal age, education, and socioeconomic status. Cumulative PM_2.5_ exposures to the complex mixture of urban ambient pollution ought to be included.Based on our research findings, APOE4 carriers living in highly polluted environments should be labeled as high risk for AD; thus, all studies should include APOE genotyping. A highly polluted environment can be defined as one with ambient pollution concentrations that exceed the existing air quality standards in the Organization for Economic Cooperation and Development [211].We can longitudinally follow BMI, BAEP changes, gait and equilibrium abnormalities, HbA1c, and dyslipidemia profiles in APOE4 subjects. Maintaining normal weight in children is imperative, and in highly polluted cities, only physical education indoor classes should be allowed at elementary, middle, and high school levels.Education efforts should be aimed at healthy lifestyle behaviors starting in infancy [214] and included in low-cost government daycares. Access to a healthy breakfast should be available in public elementary schools along with out-of-school time (OST) programs with stimulating environments, including languages, music, arts, theater, and indoor sports. OST programs must provide academic enrichment and tutorial services.Education and occupation are proxies for cognitive reserve (CR), defined by Stern [215] as “differences in cognitive processes as a function of lifetime intellectual activities or other environmental factors that explain differential susceptibility to functional impairment in the presence of pathology or other neurological insult”. Given that young adults with higher college and formal education years did significantly better at cognitive testing versus less formal education years [203] and young adulthood is associated with better late-life cognition [216], efforts should be made to keep young adults in school and increase their CR. The 2021 NEET of 18–24 y olds in Mexico must be decreased from the 22% reported [211].Good balanced diets implemented in pregnant women and in infancy are a key requirement in our Mexican populations. Two factors are important: i. healthy dietary behavior patterns and ii. monetary resources. In a country with 126.7 million people- including 55.7 million in poverty- [217] with limited access to poor medical assistance and deficient education, we have an increment in chronic non-communicable diseases, as Manderson and Jewett [218] commented, “… the result of poverty and the manipulation of food markets”. Thus, the best way to fight poverty is through the creation of local jobs, improvement in the quality of technical program training, access to education from elementary to college levels, and the retention of students at school.Purpose in life promotes resilience against brain changes already observable in middle age [219]. The work of Abellaneda-Pérez and colleagues [219] is highly recommended and emphasizes the fact that “having a purposeful life implies larger functional integration” of the dorsal default-mode network “dDMN, which may potentially reflect greater brain reserve associated with better cognitive function”. Parents and teachers could help children and teens to find purpose in their lives using a combination of experiences, education, social needs, and values.The importance of APOE isoforms on the developing brain functions and the development and progression of Alzheimer’s Disease is clear. We need to neuroprotect individuals at high risk, and certainly, APOE4 is a potential therapeutic target, as commented by Ayyubova [220]. Among the 21.8 million MMC residents, we potentially have 4.36 million APOE4 subjects at earlier accelerated AD risk. These individuals are targets for neuroprotective interventions where specific pathways can be targeted in the disease process [220].The NIA-AA framework can serve for testing the impact of particulate air pollution, specifically ultrafine PM and nanoparticles, as the key players in the processes of oxidative stress, neuroinflammation, DNA damage, protein aggregation and misfolding, and faulty complex protein quality control. We need noninvasive biomarkers indicative of the P-tau and Aβ_42_ abnormal protein deposits across the disease continuum starting in childhood. The need for non-invasive biomarkers is urgent and equally applicable to Mexican Americans, a highly vulnerable population, severely underrepresented in research [221]. There is no support for highly trained Mexican American researchers working on AD prevention.Fine particulate matter PM_2.5_, UFPs, and NPs are serious health problems in Metropolitan Mexico City [185,186,187,188,189]. The problem of particle pollution is solvable. We are knowledgeable on the impact of old heavy diesel vehicles and other sources of excessive emissions and the brain effects of combustion-generated NPs [61,62,63]. We have the technological capability and the resources to control this pollution and protect young brains. Unfortunately, we lack the political will and aptitude to acknowledge that doing nothing will be far more costly.

All these recommendations should be implemented early, knowing cognition deficits and brain structural changes start in the first two decades of life—the critical window of opportunity to implement programs to reduce PM_2.5_, UFP, and NP emissions alongside actions directed to protect children from Alzheimer’s Disease and other fatal neurodegenerative diseases. APOE4 carriers are an early high AD risk group.

Alzheimer’s disease evolving from childhood in polluted, anthropogenic, and industrial environments ought to be preventable. Neuroprotection should be our goal, and effective reductions of PM_2.5_, UFP, and NP emissions from all sources are urgently needed. Prevention ought to be at the core of the public health sector response.

## Figures and Tables

**Figure 1 biomolecules-13-00927-f001:**
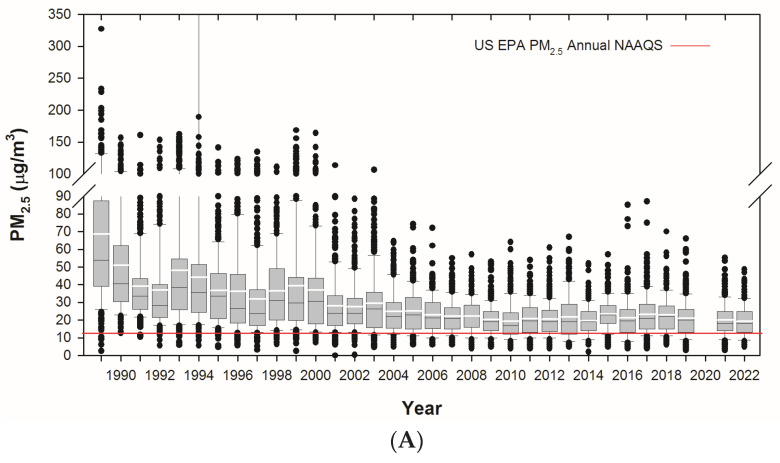
(**A**) Box and whisker plots of the trend of mean 24 h PM_2.5_ concentrations for five representative monitoring stations in MMC from 1989 to 2022 and their comparison with the respective annual US EPA NAAQS. Data were processed and evaluated from measurements reported by the manual PM network of the Secretaría del Medio Ambiente del Gobierno de la Ciudad de México (SEDEMA) under a 6-day sampling schedule. Annual means from the years before 2004 were estimated from available information on PM_10_ since 1990 and the mean slope of the correlation PM_10_ vs. PM_2.5_ between 2004 and 2007. The solid white line within the boxes represents the annual mean, and the solid black line the annual median. (**B**) Trends of estimated PNCs and the associated annual medians of 1 h average CO for five representative monitoring stations of the MMC from 1989 to 2022. The symbol # represents the number of particles. The colored circles in the figure correspond to the medians of PNCs measured by the authors referenced [186,187,188,189] in the figure and indicated by the zone in the urban area where the measurements were made. CO and PM_2.5_ data source: http://www.aire.cdmx.gob.mx/default.php# (accessed on 13 March 2023).

**Figure 2 biomolecules-13-00927-f002:**
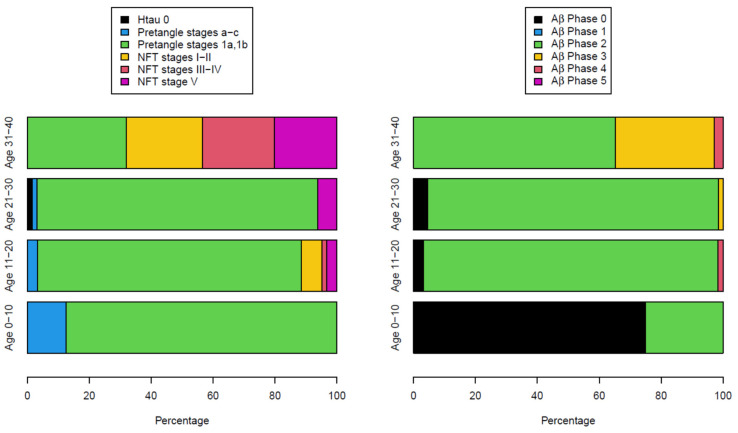
Alzheimer’s disease staging using P-tau (Htau) and Aβ_42_ in 203 autopsies [26] average age 25.4 ± 9.2 y; 202/203 had AD pathology, including the youngest subject, an 11-month-old baby.

**Figure 3 biomolecules-13-00927-f003:**
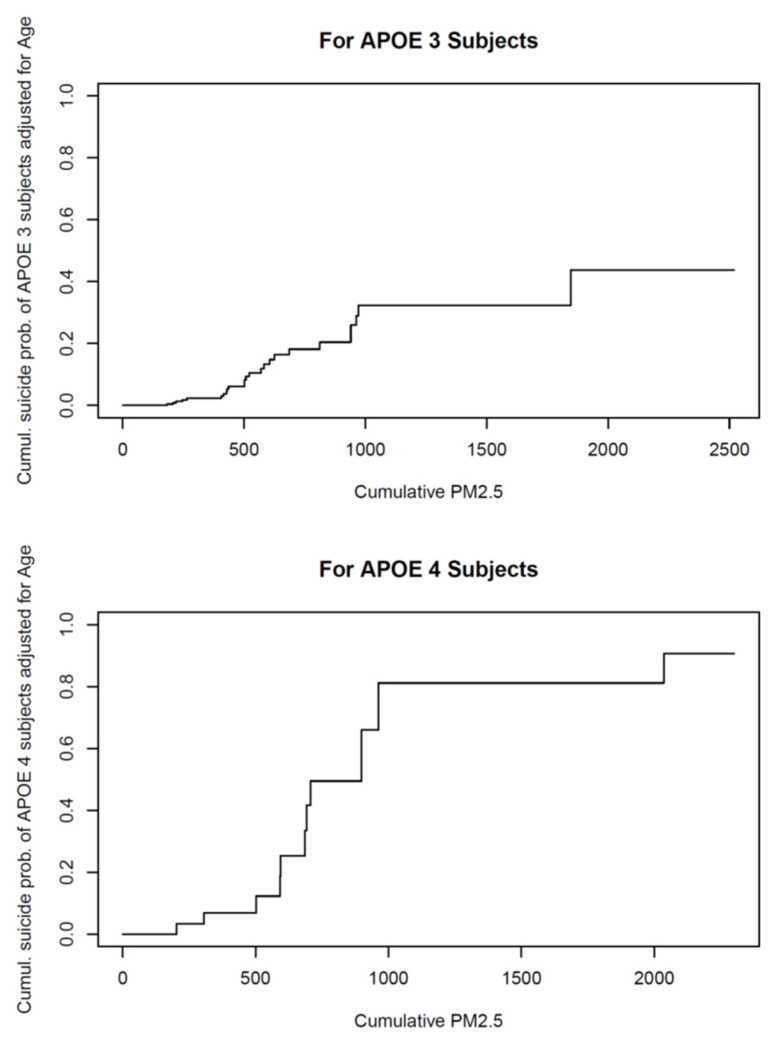
Cumulative suicide risk probability between MMC APOE3 and APOE4 subjects as a function of cumulative PM_2.5_ lifelong exposure, including pregnancy, after adjusting for age. Cox’s proportional hazard model was used to estimate the underlying survival function, and the individual’s age was used as a predictor [26].

**Figure 4 biomolecules-13-00927-f004:**
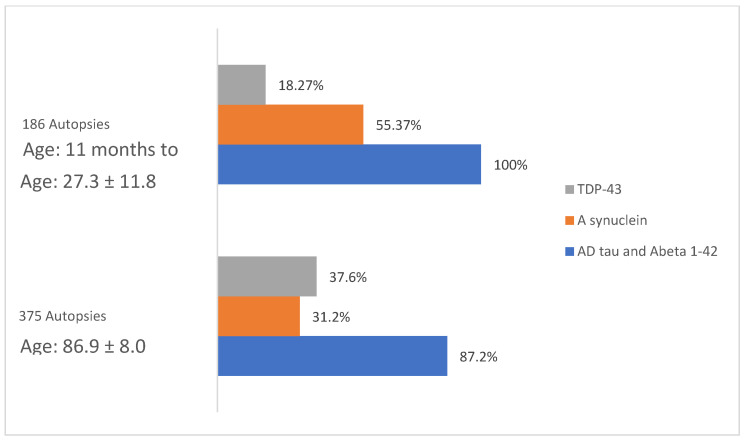
Comparison in aberrant neural proteins between the young MMC 186 autopsy cohort [27] and Karanth et al. [184]; 375 autopsies with an average age of 86.9 ± 8.04 y, including subjects with normal cognition, mild cognitive impairment (MCI), impaired (but not MCI), and dementia.

**Figure 5 biomolecules-13-00927-f005:**
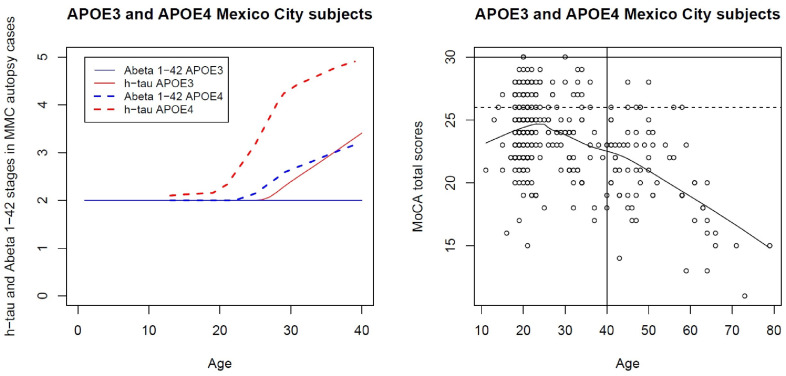
The left graphic illustrates the P-tau [82,83,84,85] and Aβ_42_ stages [212] for APOE3 and APOE4 Metropolitan Mexico City autopsy cases. The accumulation of P-tau for APOE4 carriers accelerates at the end of the second decade through the fourth, while amyloid has a slower accumulation. The risk of progressing to P-tau stage 3 or higher is significant for APOE4 subjects *p* = 0.0089, while the risk of progressing to Aβ phase 3 or higher is *p* = 0.0035, after adjusting for age and gender. The right graphic shows the average MoCA total scores for both APOE3 and APOE4; average values do not reach normal values (26–30), and after the middle of the second decade, MoCA total score goes down, coinciding with the P-tau and amyloid accumulation.

## Data Availability

Not applicable.

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
