# Peer review of "APOE Peripheral and Brain Impact: APOE4 Carriers Accelerate Their Alzheimer Continuum and Have a High Risk of Suicide in PM2.5 Polluted Cities"

_biomolecules, 2023, doi:10.3390/biom13060927_

Round 1

Author Response

Summary:

The authors present a broad literature review focused on the role of the apolipoprotein (APOE) E3 and E4 on the acceleration of the Alzheimer Continuum in children and young adults using data obtained from autopsies from Mexico City as a study population. The authors also address the role of suicide risk and the interaction of anthropogenic pollution (nanoparticles, air pollution, etc.) on APOE genotype/phenotype. The review manuscript is well-written and cites a sufficient number of references (n=220) to inform readers. The structure of the review is well considered and moves from a discussion of peripheral effects of APOE4, to the brain, to cellular components (i.e. mitochondria), then moves to clinical review on children and young adults. Data is presented here from authors previous studies and public databases to help guide them reader. I think the review is well suited for the publication in Biomolecules and recommend its publication is it will add an important road map to stimulate research in toxicological consequences of the environment on APOE3 and E4 carriers. In reading, there are a few points the authors may consider to enhance the manuscript.

Minor issues:

  • Lines 352-355: Please add reference to support this statement in text

AUTHORS RESPONSE: Thanks very much, the reference number is in place.

The authors concluded defining APOE ε polymorphisms in young children may provide the earliest indicators for individuals who might benefit from early interventions or preventive measures for future brain injuries and dementia [165]; we fully agreed with them[165].

  • Line 189: Please better define Alzheimer Continuum early in text. Its definition would better prepare the reader for the pathological data presented in the later parts of the

AUTHORS RESPONSE: Based on this suggestion, we literally changed the abstract and the Introduction, Thanks so much!  Please see below:

Abstract 

This Review emphasizes the impact of APOE4-the most significant genetic risk factor for Alzheimer's disease (AD)- upon peripheral and neural effects starting in childhood. We discuss major mechanistic players associated with APOE alleles’ in order to understand their impact from conception and through all life stages and the importance of detrimental, synergistic environmental exposures. APOE4 influences AD pathogenesis and exposures to fine particulate matter (PM2.5), ultrafine particles (UFP) associated with combustion and friction processes, and manufactured nanoparticles (NPs) appear to be major contributors to cerebrovascular dysfunction, neuroinflammation and oxidative stress. In the context of outdoor and indoor PM pollution burden and Fe, Ti, and Al alloys, Hg, Cu, Ca, Sn, and Si NPs in placenta and fetal brain tissues, urban APOE3 and 4 carriers are developing AD biological disease hallmarks (hyperphosphorylated tau P-tau and amyloid beta 42 plaques Aβ42 ). Strikingly, young ≤40y APOE4 carriers are having 4.92 times higher suicide odds and 23.6 times higher odds of reaching Braak NFT V stages versus APOE4 non-carriers.

The National Institute on Aging and Alzheimer’s Association NIA-AA framework can serve to test the hypothesis UFP and NPs are key players for oxidative stress, neuroinflammation, protein aggregation and misfolding, faulty complex protein quality control and cell membranes and organelle early damage to neural and vascular cells. Noninvasive biomarkers indicative of the P-tau and Aβ42 abnormal protein deposits are needed across the disease continuum starting in childhood. Among the 21.8 million MMC residents, we have potentially 4 million APOE4 carriers at accelerated AD progression. These APOE4 individuals are prime candidates for early neuroprotective interventional trials.

APOE4 is key in the development of AD evolving from childhood in highly polluted, urban centres dominated by anthropogenic and industrial sources of pollution. APOE4 subjects are at higher risk of AD development and neuroprotection ought to be implemented. Effective reductions of fine particulate matter (PM2.5), UFP and NPs emissions from all sources are urgently needed. AD prevention ought to be at the core of the public health sector response.

Keywords: air pollution, children, Alzheimer, APOE4, early biomarkers, amyloid beta, cognition, hyperphosphorylated tau, Metropolitan Mexico City, nanoparticles, neuroprotection, ultra fine particulate matter, PM2.5

1.Introduction

Growing body of evidence indicates the strong association between Apolipoprotein APOE4 allele and increased risk of Alzheimer’s disease (AD), the associations with cognitive and brain volumetric changes from childhood and the increased risk for other neurodegenerative diseases besides AD[1-18].It has been almost five decades since Shore and Shore [19] landmark paper on the heterogeneity of human plasma very low density lipoproteins and the current APOE allele differential impacting across global populations [20-23]. APOE4 carrying populations were protected through evolutionary pressure against infectious diseases, including malaria, thus explaining the prevalence of alleles differences across races and ethnicity [23]. Corvo et al., [20] discussed a very critical aspect of APOE allele distribution; APOE3 is the most frequent allele regardless of race and ethnicity, associated with populations with long-established agricultural economies i.e., the Mediterranean basin, while APOE4, the ancestral allele, is higher in pygmies, Malaysia aborigines, and Native Americans with economy’s characteristic of foraging and/or food scarcity. Among Latinos in the USA [24], Caribbean Latinos from the Dominican Republic and Puerto Rico have the highest APOE4 frequency, both in normal cognitive subjects 23.2% versus AD patients 32.4%, while Mexican Hispanic AD cases versus controls recorded: 21.4% and 12.5%, respectively [25].The world APOE4 frequency varies from 9%-23% i.e., Asian 9%, Hispanic 12%, white 14%, African descent 19%, other/mixed 23%) and dramatically increases in AD patients (Hispanic 24%, Asian 28%, African descent 35%, white 38%, other/mixed 45%)[21].

This Review emphasizes the impact of APOE4 peripheral and neural consequences starting in childhood, the environmental factors appearing to synergistically damage neural and extraneural key systems and impacting the development and progression of AD and other fatal neurodegenerative diseases [26-29]. We discuss the APOE4 effects that influence AD pathogenesis and how exposures to ultrafine particles (UFP) associated with fossil fuel combustion and friction processes and industrial manufactured nanoparticles (NPs) are major contributors to an ample variety of neural effects, including cerebrovascular dysfunction, neuroinflammation, oxidative stress and extensive organelle pathology and dysfunction [30-42]. We highlight hepatocytes as one of the largest contributors to the peripheral APOE pool, making the liver, an organ with significant importance at times where we are facing serious liver metabolic pathologies in children and young adults [43-45].

Understanding the extensive APOE-associated detrimental neural and nonneural effects right from conception and through all life stages and the importance of exposures to environmental aggressors allows the understanding of why an 11-year old has extensive hyperphosphorylated tau in brainstem and cortical locations; why we have diffuse beta amyloid plaques in infants, and why APOE4 children and young adult carriers are committing suicide and accelerating their neurodegenerative processes in highly polluted environments. The challenges we face dealing with pediatric and young adult highly exposed PM populations,particularly APOE4 carriers, include i. Early identification of high risk AD young subjects, including children, ii. Making an early AD diagnosis using non-invasive biomarkers indicative of neuropathological progressive changes, iii. Defining and staging the disease from pediatric ages across the heterogeneity of the AD spectrum, and iv. Protecting millions of exposed people across the world. Alzheimer’s disease associated to air pollution ought to be preventable.

  • Although data is observational in nature, it would be informative to know how these findings compare to children and young adults in other cities in Mexico with better social economic status and less recorded

AUTHORS RESPONSE: yes, we have cognitive studies, BAEPs, olfaction and MRI from low polluted children and adults with normal parameters and measurements versus Mexico City residents. We also have control low pollution autopsies matched with MMC residents age, lacking neurodegenerative changes.

  • APOE2 genotype, which is protective from AD, is only discussed briefly (Lines 247- 248). A few sentences regarding APOE2 and its frequency and its impact in the Mexico City population would be very informative.
  • AUTHORS RESPONSE: There is a good reason why we do not have APOE2 subjects in Mexico. We are mostly a mixture of native indigenous populations and Spaniards, I have two APOE 2/3 in more than 1200 genotyped cases in my lab (alive and autopsies).
  • Line 354: Figure is pixeled, please add figure with higher resolution
  • Line 743: Figure is pixeled, please add figure with higher resolution

AUTHORS RESPONSE: Done

I recommend several minor edits to help with presentation. Note: (>) symbolizes change to:

  • Please select a uniform text for It is presented several ways in the manuscript (e.g. APOE4, Apoe4, APOE e4, etc.)

AUTHORS RESPONSE:  Thanks so much! Yes, we talk about APOE genes and ApoE protein

APOE polymorphic alleles directly influence plasma and brain cholesterol concentrations [43,50,51] and the liver is key for Apolipoprotein E (ApoE) -the cholesterol carrier, key for lipid transport and injury repair in the brain- reuptake,

    • Line 5: 3sd > 3rd
    • Line 71 & 72: mixed > mixed race
    • Line 190: Please spell out MCI where it first appears in the manuscript
    • Line 211: Sporadic AD (sAD)
    • Line 299: Please spell out LPS where it first appears in the manuscript1

  • Please select a uniform text for numbering: Some numbers are presented with comma and others are not (e.g. 1000 or 1,000) See line 336 for an example.
  • Line 344: African Americans > AA
  • Line 350: Non-Hispanic Whites (NHW)
  • Lines 353-355: Please add reference for this sentence
  • Lines 382, 392, 437: ug/m3 to ug/m3
  • Line 390: Salinas CA > Salinas, CA USA
  • Line 400: Please spell out SES where it first appears in the manuscript
  • Lins 471 & 483: 31Phosphorus > 31Phosphorus
  • Lines 473: Alzheimer’s disease patients
  • Line 540: define size of 5
  • Line 545: Figure 1B: please cite source of data
  • Line 575: Figure 2: please cite source of data
  • Line 585: Figure 3: please cite source of data
  • Line 591: Figure 2: Please spell out FTLD, LBD, PD, and ALS
  • Line 625: TDP 43 > TDP-43
  • Line 626: Please spell out CSF where it first appears in the manuscript
  • Line 627 Please spell out Aβ1-42 where it first appears in the manuscript
  • Line 709: Please spell out MoCA where it first appears in the manuscript
  • Line 724: : Please spell out NIA-AA where it first appears in the manuscript

DONE

THANKS SO MUCH FOR YOUR INPUT, IT WAS VERY HELPFUL TO IMPROVE OUR WORK.

Reviewer 2 Report

This is a comprehensive review with 220 bibliographic references on the role of APOE in AD.  The authors postulate that in Metropolitan Mexico City pollutant nanoparticles begin affecting the brain in the first year of life and continue until late in life when the clinical manifestations of AD become clinically eloquent.

The main problem with this review is that the data provided is based mainly on epidemiological observations and it is well-known that “association is not causation” until proven otherwise.  For the readers of this review to accept the argument that EP causes AD and therefore to conclude that AD could be potentially preventable the authors must provide comprehensive data from animal studies (i.e., in mice by Ehsanifar et al Neurochem Int 2021 and Antsiferova et al Toxics 2021; dogs by Calderón-Gracidueñas et al) as well as from human brain studies.  A detailed description of the animal models will be required as well as a condensation in a Table of the information on the effects of NP in the brain of animals.  This information will provide a strong argument.

The next step is to provide a comprehensive description of the neuropathological data collected by the authors [26].  A description of the histopathology and immunohistochemistry methods used to confirm presence of tau protein and beta amyloid with images should be provided.  Again, in addition to Figure 2, a Table summarizing the findings of the 202/203 (how many?) autopsies studied by the authors in human subjects ages 11 months to 40 y/o will be a very strong argument. Images of the amyloid and tau neuropathology observed in the 11-year old with extensive hyperphosphorylated tau in brainstem and cortical locations; of the diffuse beta amyloid plaques in infants, and of the accelerated neurodegenerative processes in young people living in highly polluted environments who committed suicide will be extremely helpful for the authors’ goals.  Once the animal and human neuropathology data are provided to the reader, then the pathogenesis of AD related to APOE can be explained.

The authors should mention that selenium and other nanoparticles have been proposed as treatment for AD (Martin-Rapun et al., Curr Pharm Design 2017).

There is no need to give the first and last names of the authors (i.e., Shore VG and Shore B, 1973; Ana B Martínez-Martínez and colleagues [49]).  Last name and number of the bibliographic reference suffice.

Author Response

Review Report Form REVIEWER 2

AUTHORS COMMENTS

Thank you so much for your comments and observations.

The biological diagnosis of AD is very clear in the literature:  Amyloid β plaques and hyperphosphorylated tau abnormal proteins define AD as a unique neurodegenerative disease [181].

  1. Jack, C. R.; Bennett, D. A.; Blennow, K.; Carrillo, M. C.; Dunn, B.; Haeberlein, S. B.; Holtzman, D. M.; Jagust, W.; Jessen, F.; Karlawish, J.; Liu, E.; Molinuevo, J. L.; Montine, T.; Phelps, C.; Rankin, K. P.; Rowe, C. C.; Scheltens, P.; Siemers, E.; Snyder, H. M.; Sperling, R.; Contributors. NIA-AA Research Framework: Toward a Biological Definition of Alzheimer’s Disease. Alzheimers Dement 2018, 14 (4), 535–562. https://doi.org/10.1016/j.jalz.2018.02.018

  1. Calderón-Garcidueñas, L.; Gónzalez-Maciel, A.; Reynoso-Robles, R.; Delgado-Chávez, R.; Mukherjee, P. S.; Kulesza, R. J.; Torres-Jardón, R.; Ávila-Ramírez, J.; Villarreal-Ríos, R. Hallmarks of Alzheimer Disease Are Evolving Relentlessly in Metropolitan Mexico City Infants, Children and Young Adults. APOE4 Carriers Have Higher Suicide Risk and Higher Odds of Reaching NFT Stage V at ≤ 40 Years of Age. Environ Res 2018, 164, 475–487. https://doi.org/10.1016/j.envres.2018.03.023.

These 202 of 203 forensic autopsy results of subjects between 11 months and 40 years of age dying suddenly [26], fulfilled the biological definition of AD and their alive and seemingly healthy MMC counterparts have cognitive deficits, low CSF Aβ 42, brain atrophy by MRI, etc., thus the preclinical time is short and  eloquent from age 10, when we pick up the cognitive significant differences between high and low pollution and of course cognition decreases progressively from the midst of the second decade and over [Figure 5].

The main problem with this review is that the data provided is based mainly on epidemiological observations and it is well-known that “association is not causation” until proven otherwise.  For the readers of this review to accept the argument that EP causes AD and therefore to conclude that AD could be potentially preventable the authors must provide comprehensive data from animal studies (i.e., in mice by Ehsanifar et al Neurochem Int 2021 and Antsiferova et al Toxics 2021; dogs by Calderón-Gracidueñas et al) as well as from human brain studies.  A detailed description of the animal models will be required as well as a condensation in a Table of the information on the effects of NP in the brain of animals.  This information will provide a strong argument.

 AUTHORS COMMENTS

This APOE Review included epidemiological and clinical and imagen studies of APOE4 versus APOE3 and 2 subjects with a focus on peripheral and brain effects.

Yes, we cited the work by Ehsanifar et al. [62] whom in fact are one of the best literature sources of NPs effects on animal studies.

  1. Ehsanifar, M.; Montazeri, Z.; Taheri, M. A.; Rafati, M.; Behjati, M.; Karimian, M. Hippocampal Inflammation and Oxidative Stress Following Exposure to Diesel Exhaust Nanoparticles in Male and Female Mice. Neurochem Int 2021, 145, 104989. https://doi.org/10.1016/j.neuint.2021.104989.

You are correct, animals, especially dogs are a very good model for AD development, we have published in the area.

The next step is to provide a comprehensive description of the neuropathological data collected by the authors [26].  A description of the histopathology and immunohistochemistry methods used to confirm presence of tau protein and beta amyloid with images should be provided.  Again, in addition to Figure 2, a Table summarizing the findings of the 202/203 (how many?) autopsies studied by the authors in human subjects ages 11 months to 40 y/o will be a very strong argument. Images of the amyloid and tau neuropathology observed in the 11-year old with extensive hyperphosphorylated tau in brainstem and cortical locations; of the diffuse beta amyloid plaques in infants, and of the accelerated neurodegenerative processes in young people living in highly polluted environments who committed suicide will be extremely helpful for the authors’ goals.  Once the animal and human neuropathology data are provided to the reader, then the pathogenesis of AD related to APOE can be explained.

AUTHORS COMMENTS

As with every review paper, we are citing already published literature. We condensed the information from each paper and so i.e., a Table summarizing the findings of the 202/203 (how many?) autopsies is indeed in the original paper of 203 forensic autopsies, 202 of which we could stage for both P-tau and Aβ 42 [26]. We followed the sequence of papers focusing on peripheral and central effects of APOE alleles, the clinical and imagen studies and Figures 2-5 illustrated what we have documented in a megacity (including the forensic data).

The authors should mention that selenium and other nanoparticles have been proposed as treatment for AD (Martin-Rapun et al., Curr Pharm Design 2017).

AUTHORS COMMENTS

Yes, amazing paper, we are familiar with them, however the focus of the paper was no treatment, that will be a very different work for the future.

Comments on the Quality of English Language

There is no need to give the first and last names of the authors (i.e., Shore VG and Shore B, 1973; Ana B Martínez-Martínez and colleagues [49]).  Last name and number of the bibliographic reference suffice.

AUTHORS COMMENTS

We have changed that, thanks so much.

The authors thank this Reviewer for his Comments.

Submission Date

07 April 2023

Date of this review

Reviewer 3 Report

Dear authors, the paper is very specific, with several citations and technical references. Authors deal with a thorny argument, potentially with significant impact on clinical research.

However, I noticed some points in the manuscript that deserve to be ameliorated.

1) The title is too long and in this form is less understandable. Please change. I also suggest to reformulate the abstract and the various key arguments treated in the paper in a different manner along the text, in order to link in a more linear way the single points (see comments below).

2) Abstract: the first sentence appears very focused on Alzheimer's disease, so I suggest to attenuate the formulation; maybe it can be suitable to build two different sentences, one relative to APOE peripheral and neural effects in infancy and one focused on the role for AD development.

3) Please ascertain that all acronims in the text are accompanied by previous definition in extense (i.e., NIA-AA, MCI, FA). Please change Continuum with continuum.

4) Abstract: authors deal with APOE impact on suicide odds. I less understand how the previous sentence is linked to this finding, I believe it deserves to be further explained, in order to not give impression of a direct link with suicide risk for all subjects affected by APOE3 and 4 alleles, if I well understand the intent. 

5) Introduction: authors report that APOE is associated with dementia; APOE is clearly associated with brain amyloidosis, but the link with dementia seems to not be direct. I suggest to further discuss this point. I also suggest to explain the different implications of allele 3 and 4 for AD.

6) I suggest to move some sentences,  that confer a summary picture of the topic, in other points of the text: line 155-169 pag. 4 to be moved previously; line 371-374 pag. 8 to be moved previously. Conversely, I suggest to move the sentence line 538-543 pag. 11 to line 570 pag. 12.

7) I suggest to reformulate the following sentences, by creating more brief sentences and by a revision of punctuation:

line 124-128 pag. 3, line132-135 pag. 3, line 143-147 pag. 3 (the sentence should be better explained; also replace "are" with "were"), line 255-260 pag. 6, line 487-492 pag. 10; line 626-627 pag. 15.

8) line 224-225: please better explain the finding of Trem2 deficiency for neurodegeneration.

9) line 498-499: I suggest to reformulate the sentence in which authors state that female sex is a risk factor; gender per se is not a risk factor, even if it is more strongly associated.

10) in two different parts of the text, authors should put attention to reformulate, especially the citations: line 578-583, in particular specify with accurate revision what results are linked to MMC study and what are found in other research (i.e., cit 191 refers to Braak); paragraph 4.1: I suggest to better explain the link between APOE findings and pollution impact on APOE. Maybe authors could at first deal with APOE in children and young adults, and then focus on the pollution impact.

11) Line 242: replace "participated" with "took place"; line 250: add "that"; line 248: add "respectively"; line 268: change "related to direct" with "are directly related"; line 270: change the term ("key" with "key driver" or "key feature"); line 306: change "coincide in" with "agree about"; line 493: specify that the conclusions are related to AD condition; line 614: change "nigrae" with "nigra".

12) The NIA-AA framework is mainly a research framework. Authors correctly report some insufficiency of this model to take into account some environmental and other factors on AD pathology.  However, I suggest to extend this critical point to other models of neurodegeneration. In fact, the point underlined by authors derserve attention in the public policies, and in the framework of health prevention and promotion. 

The paper is very technical and well-documented. However, as suggested in the comments, I strongly recommend a deep revision of English language, since in some points of the manuscript I consider difficult to understand what authors are explaining.

Author Response

REVIEWER #3

Dear authors, the paper is very specific, with several citations and technical references. Authors deal with a thorny argument, potentially with significant impact on clinical research.

However, I noticed some points in the manuscript that deserve to be ameliorated.

  • The title is too long and in this form is less understandable. Please change. I also suggest to reformulate the abstract and the various key arguments treated in the paper in a different manner along the text, in order to link in a more linear way the single points (see comments below).

AUTHORS COMMENTS

Thanks very much! We have changed the title. The new title reads:

APOE peripheral and brain impact. APOE4 carriers accelerate their Alzheimer Continuum and have a high risk of suicide in highly polluted cities. Control of anthropogenic and industrial nanoparticle emissions and early neuroprotection are urgent.

  • Abstract: the first sentence appears very focused on Alzheimer's disease, so I suggest to attenuate the formulation; maybe it can be suitable to build two different sentences, one relative to APOE peripheral and neural effects in infancy and one focused on the role for AD development.

AUTHORS COMMENTS

Thanks so much! We changed almost the entire abstract, here it is:

This Review emphasizes the impact of APOE4-the most significant genetic risk factor for Alzheimer's disease (AD)- upon peripheral and neural effects starting in childhood. We discuss major mechanistic players associated with APOE alleles’ in order to understand their impact from conception and through all life stages and the importance of detrimental, synergistic environmental exposures. APOE4 influences AD pathogenesis and exposures to fine particulate matter (PM2.5), ultrafine particles (UFP) associated with combustion and friction processes, and manufactured nanoparticles (NPs) appear to be major contributors to cerebrovascular dysfunction, neuroinflammation and oxidative stress. In the context of outdoor and indoor PM pollution burden and Fe, Ti, and Al alloys, Hg, Cu, Ca, Sn, and Si NPs in placenta and fetal brain tissues, urban APOE3 and 4 carriers are developing AD biological disease hallmarks (hyperphosphorylated tau P-tau and amyloid beta 42 plaques Aβ42 ). Strikingly, young ≤40y APOE4 carriers are having 4.92 times higher suicide odds and 23.6 times higher odds of reaching Braak NFT V stages versus APOE4 non-carriers.

The National Institute on Aging and Alzheimer’s Association NIA-AA framework can serve to test the hypothesis UFP and NPs are key players for oxidative stress, neuroinflammation, protein aggregation and misfolding, faulty complex protein quality control and cell membranes and organelle early damage to neural and vascular cells. Noninvasive biomarkers indicative of the P-tau and Aβ42 abnormal protein deposits are needed across the disease continuum starting in childhood. Among the 21.8 million MMC residents, we have potentially 4 million APOE4 carriers at accelerated AD progression. These APOE4 individuals are prime candidates for early neuroprotective interventional trials.

APOE4 is key in the development of AD evolving from childhood in highly polluted, urban centres dominated by anthropogenic and industrial sources of pollution. APOE4 subjects are at higher risk of AD development and neuroprotection ought to be implemented. Effective reductions of fine particulate matter (PM2.5), UFP and NPs emissions from all sources are urgently needed. AD prevention ought to be at the core of the public health sector response.

  • Please ascertain that all acronims in the text are accompanied by previous definition in extense (i.e., NIA-AA, MCI, FA). Please change Continuum with continuum.

AUTHORS COMMENT

We defined all acronyms, however in the current literature Continuum is written with capital C

  • Abstract: authors deal with APOE impact on suicide odds. I less understand how the previous sentence is linked to this finding, I believe it deserves to be further explained, in order to not give impression of a direct link with suicide risk for all subjects affected by APOE3 and 4 alleles, if I well understand the intent. 

AUTHORS COMMENT

We changed the entire paragraph. Please see above #2

  • Introduction: authors report that APOE is associated with dementia; APOE is clearly associated with brain amyloidosis, but the link with dementia seems to not be direct. I suggest to further discuss this point. I also suggest to explain the different implications of allele 3 and 4 for AD.

AUTHORS COMMENT  We changed the Introduction ?

1.Introduction

Growing body of evidence indicates the strong association between Apolipoprotein APOE4 allele and increased risk of Alzheimer’s disease (AD), the associations with cognitive and brain volumetric changes from childhood and the increased risk for other neurodegenerative diseases besides AD[1-18].It has been almost five decades since Shore and Shore [19] landmark paper on the heterogeneity of human plasma very low density lipoproteins and the current APOE allele differential impacting across global populations [20-23]. APOE4 carrying populations were protected through evolutionary pressure against infectious diseases, including malaria, thus explaining the prevalence of alleles differences across races and ethnicity [23]. Corvo et al., [20] discussed a very critical aspect of APOE allele distribution; APOE3 is the most frequent allele regardless of race and ethnicity, associated with populations with long-established agricultural economies i.e., the Mediterranean basin, while APOE4, the ancestral allele, is higher in pygmies, Malaysia aborigines, and Native Americans with economy’s characteristic of foraging and/or food scarcity. Among Latinos in the USA [24], Caribbean Latinos from the Dominican Republic and Puerto Rico have the highest APOE4 frequency, both in normal cognitive subjects 23.2% versus AD patients 32.4%, while Mexican Hispanic AD cases versus controls recorded: 21.4% and 12.5%, respectively [25].The world APOE4 frequency varies from 9%-23% i.e., Asian 9%, Hispanic 12%, white 14%, African descent 19%, other/mixed race 23%) and dramatically increases in AD patients (Hispanic 24%, Asian 28%, African descent 35%, white 38%, other/mixed race 45%)[21].

This Review emphasizes the impact of APOE4 peripheral and neural consequences starting in childhood, the environmental factors appearing to synergistically damage neural and extraneural key systems and impacting the development and progression of AD and other fatal neurodegenerative diseases [26-29]. We discuss the APOE4 effects that influence AD pathogenesis and how exposures to fine particulate matter (PM2.5, particles ≤2.5 μm), ultrafine particles (UFP) associated with fossil fuel combustion and friction processes and industrial manufactured nanoparticles (NPs) are major contributors to an ample variety of neural effects, including cerebrovascular dysfunction, neuroinflammation, oxidative stress and extensive organelle pathology and dysfunction [30-42]. We highlight hepatocytes as one of the largest contributors to the peripheral APOE pool, making the liver, an organ with significant importance at times where we are facing serious liver metabolic pathologies in children and young adults [43-45].

Understanding the extensive APOE-associated detrimental neural and nonneural effects right from conception and through all life stages and the importance of exposures to environmental aggressors allows the understanding of why an 11-year old has extensive hyperphosphorylated tau in brainstem and cortical locations; why we have diffuse beta amyloid plaques in infants, and why APOE4 children and young adult carriers are committing suicide and accelerating their neurodegenerative processes in highly polluted environments. The challenges we face dealing with pediatric and young adult highly exposed PM populations,particularly APOE4 carriers, includes i. Early identification of high risk AD young subjects, including children, ii. Making an early AD diagnosis using non-invasive biomarkers indicative of neuropathological progressive changes, iii. Defining and staging the disease from pediatric ages across the heterogeneity of the AD spectrum, and iv. Protecting millions of exposed people across the world. Alzheimer’s disease associated to air pollution ought to be preventable.

6) I suggest to move some sentences,  that confer a summary picture of the topic, in other points of the text: line 155-169 pag. 4 to be moved previously; line 371-374 pag. 8 to be moved previously. Conversely, I suggest to move the sentence line 538-543 pag. 11 to line 570 pag. 12.

7) I suggest to reformulate the following sentences, by creating more brief sentences and by a revision of punctuation:

line 124-128 pag. 3, line132-135 pag. 3, line 143-147 pag. 3 (the sentence should be better explained; also replace "are" with "were"), line 255-260 pag. 6, line 487-492 pag. 10; line 626-627 pag. 15.

AUTHORS COMMENT:

We  made a lot of changes throughout the paper, including your suggestions, they are marked in yellow in the highlighted version.

8) line 224-225: please better explain the finding of Trem2 deficiency for neurodegeneration.

AUTHORS COMMENT:

Absolutely:

Neuroinflammation plays an important role in AD and variants in APOE and the triggering receptor expressed on myeloid cells 2 (TREM2)-a microglial receptor-adaptor complex, expressed on myeloid cells 2 and which three domains are key for the binding of pathogen-associated pattern molecules PAMPS (the release of protective molecules for microglial survival and the propagation of proinflammatory signals) are linked to APOE [91-93]. Gratuze et al., [92] explored whether TREM2 KO (T2KO) would block neurodegeneration in P301S Tau mice expressing ApoE4 (TE4), in a matter similar to microglial depletion. Instead of the expected decreased neurodegeneration and tau pathology in TE4-T2KO, the researchers found TREM2 deficiency aggravates neurodegeneration and concluded TREM2-independent microgliosis, facilitates tau-mediated neurodegeneration in the presence of APOE4.Heneka [93] in his Comment of Gratuze et al work [92] makes a practical key point: APOE effects have much to do with its interactions with other genetic factors, in this case TREM2 or as in the work of Vontell et al., [94] the inflammasome sensor proteins NOD-like receptor proteins (NLRP) 1 and 3, and caspase-1 are crucial in the early pathological stages of AD; thus studying only targeted genetic risk factors for sAD indeed do not give us the full picture.

9) line 498-499: I suggest to reformulate the sentence in which authors state that female sex is a risk factor; gender per se is not a risk factor, even if it is more strongly associated.

[172]. Background: The current pilot study was designed to examine the association between hippocampal γ-aminobutyric acid (GABA) concentration and episodic memory in older individuals, as well as the impact of two major risk factors for Alzheimer's disease (AD)-female sex and Apolipoprotein ε4 (ApoE ε4) genotype-on this relationship. 

Female sex is a risk factor for AD (very unfortunately for women ☹). So, the sentence reads:

Female sex is a powerful risk factor for AD and thus moderates the relationship between hippocampal GABA+ and episodic memory resulting in women having lower GABA+ concentration and worse memory performance [172]. These findings in a small sample size have no association with APOE4 [172].

I have no clue how to reformulate the sentence.

10) in two different parts of the text, authors should put attention to reformulate, especially the citations: line 578-583, in particular specify with accurate revision what results are linked to MMC study and what are found in other research (i.e., cit 191 refers to Braak); paragraph 4.1: I suggest to better explain the link between APOE findings and pollution impact on APOE. Maybe authors could at first deal with APOE in children and young adults, and then focus on the pollution impact.

AUTHORS COMMENT:

It is extremely difficult for somebody living away from a megacity where millions of people have been steadily exposed to high concentrations of PM and a myriad of other pollutants, to follow and grasp the severity of the health crisis we are witnessing. Our reality is that we deal everyday with cognitively impaired people, students struggling in the classrooms, parents unable to cope with behavioral disorders, high levels of aggression in the north part of the city (where the highest levels of PM2.5 and NPs are documented).  The “good” thing about pathology is that you have the slides, the paraffin block can be cut ≥100 times and you will still see the neurofibrillary tangles and the amyloid plaques and then check the age of the individual, a child or a young adult and you will go through the pain of seeing case after case (202 to be exact ≤40y with the AD pathology.

So, the way we put the chapters, starting with peripheral and then neural APOE effects, followed by what we see in Mexico City was the best approach. Truly, we have been documenting for years CSF, BAEPs, olfactory and gait and balance abnormalities and of course cognition in these seemingly healthy subjects…….there is nobody healthy and the 25% of the population cognitively intact, are mostly college and beyond, educated people.

The neuropath in young people is very similar to individuals 80y old [184] (Karanth et al., 2020), except we are seeing it 50-60 years earlier. And the P-tau goes hand in hand with the cognition deficits in young adults (Figure 5).

Let me tell you what the saddest of this is: nobody is interested in prevention, they want a pill that returns the brain from 850gr to 1280 gr, there is no such thing, we are seeing the slow development of AD in children, once the P-tau is in place, there is no return(ps Dr.Braak, my favorite neuropathologist, said that)

We have asked for grant support, nothing, in spite of extensive training, publications and hard work, so this is one more paper showing the extent of the serious problem, 75% of the population with various degrees of cognition problem plus, plus, plus. I get 500-700 US /year for research if I am lucky.

  1. APOE, Alzheimer’s disease, air pollution and nanoparticles. The impact of air pollution on the progression of biological Alzheimer in pediatric and young adult Metropolitan Mexico City residents.

6.CSF Alzheimer and TDP 43 pathology markers in MMC children and young adults

7.Cognition deficits in MMC APOE4 children are significantly higher than APOE3

8.Summary

11) Line 242: replace "participated" with "took place"; line 250: add "that"; line 248: add "respectively"; line 268: change "related to direct" with "are directly related"; line 270: change the term ("key" with "key driver" or "key feature"); line 306: change "coincide in" with "agree about"; line 493: specify that the conclusions are related to AD condition; line 614: change "nigrae" with "nigra".

AUTHORS COMMENT:

Done

12) The NIA-AA framework is mainly a research framework. Authors correctly report some insufficiency of this model to take into account some environmental and other factors on AD pathology.  However, I suggest to extend this critical point to other models of neurodegeneration. In fact, the point underlined by authors derserve attention in the public policies, and in the framework of health prevention and promotion. 

AUTHORS COMMENT

We are very aware of the limitation of the NIA-AA in the setting of severe air pollution and AD developing in youngsters (we are also aware of their denial of the epidemiological air pollution associations with AD and worse, if you have amyloid and tangles at age 14y you do not have AD ????? so their biological definition of biological AD is -somehow-not applicable to children or 20 y olds…………..) amyloid plaques and NFT deposits define AD as a unique neurodegenerative disease  [181].

Thus, we added a whole paragraph in the Introduction:

The challenges we face dealing with pediatric and young adult highly exposed PM populations,particularly APOE4 carriers, includes i. Early identification of high risk AD young subjects, including children, ii. Making an early AD diagnosis using non-invasive biomarkers indicative of neuropathological progressive changes, iii. Defining and staging the disease from pediatric ages across the heterogeneity of the AD spectrum, and iv. Protecting millions of exposed people across the world. Alzheimer’s disease associated to air pollution ought to be preventable.

Made emphasis is the OVERLAP of AD, PD and TDP-43 observed in the young MMC residents versus the much older individuals (there is a common denominator that keeps going to the old age).

We have described the remarkable AD, PD and TDP-43 pathology overlap in young MMC residents and the fact that is very similar to the mixed protein pathologies in elderly patients with AD, frontotemporal lobar dementia (FTLD), Lewy body disease (LBD), Parkinson’s disease (PD), amyotrophic lateral sclerosis (ALS), cerebral amyloid angiopathy (CAA), and in younger than 60y AD demented patients [180,184, 192-196].  Hyperphosphorylated tau is the major aberrant protein in MMC young residents and P-tau in substantia nigra and lack of nuclear TDP-43 in cortical motor neurons, motor neurons for cranial nerves III, V and XII and cervical motor neurons in teens and young adults were common findings [26-28].  Figure 4 shows the aberrant neural protein overlap in MMC 186 young residents compared [27] to elderly subjects in the work of 375 autopsies done by Karanth et al [184].

And end with a series of recommendations, that mostly will be ignored, certainly by the authorities in Mexico (yesterday April 29,2023 they destroyed the entire research system in the country   https://heraldodemexico.com.mx/opinion/2023/4/29/los-nuevos-amos-de-la-ciencia-en-mexico-501609.html), so no hope of one single cent going to this type of research.

The revised Summary goes over our concerns in terms of the NIA-AA research framework in the setting of severe air pollution and evaluation of young exposed populations.

Summary

APOE4 is the largest genetic risk factor for sporadic Alzheimer’s disease (sAD) and has significant detrimental impact in children and young adults subject to sustained PM2.5 and NPs exposures [26-29, 36-41]

We have shown the presence of P-tau and Aβ42 in Metropolitan Mexico City (MMC) 202 of 203 forensic autopsies random sampling in the course of 5 years, in subjects aged 11 months to 40y with no major extra neural pathology. The AD hallmarks are already present in MMC infants and progression is seen as the subjects grow up in the complex mixture of toxic pollutants. The cognition changes are documented in the first two decades of life, along with brain structural changes, abnormal gait and equilibrium and BAEPs, supported by auditory nuclei dysmorphology, brainstem accumulation of aberrant neural proteins and high concentration of magnetic NPs in the atrophic cerebellum. The overlap of P-tau, Aβ 42, α-synuclein and TDP-43 is documented in the 202 autopsies and the brain MRI in young seemingly healthy volunteers is showing cortical and subcortical atrophy overlapping AD, PD and FTLD patterns, including a striking caudate and cerebellar atrophy.

These quadruple pathologies affect both APOE3 and 4 MMC subjects and noteworthy, the progression to AD accelerates in APOE4 adults by the beginning of the third decade of life [26,101]. Overweight APOE4 girls are at high risk of metabolic syndrome, with IQ losses of ≥10 points [141,142]. Young APOE4 males are at high risk for suicide over their APOE3 counterparts [26].

A major concern in children and young adults is their subjective memory complaints and their related academic problems at school in keeping with their combination of memory, executive functions and fluid cognition deficits documented in 10.7±2.7y MMC healthy children vs controls [36,101,142,157] and Montreal Cognitive Assessment (MoCA) in the mild cognitive impairment (MCI) and dementia ranges in young adults [201,202].

In sharp contrast with the management of elderly subjects, in MMC we are dealing with a young population with progressive cognitive deficits, unable to achieve optimal performance at all levels of education. Although there is a significant cognitive advantage for those young adults with the most years of formal education, cognition deficits and/or MRI atrophy are documented in MMC healthy young volunteers regardless of APOE status. This observation has a high impact in a country where only a reduced number of students reach bachelor’s degrees [207] and has the lowest tertiary attainment rate (27%) among 25-34 olds in the Organization for Economic Cooperation and Development (OECD) [208]. In Mexico, only 31% of individuals reported having basic ICT skills (i.e., activities such as knowing how to send an email with an attachment), which is less than the OECD average (55%). Mexican adults 18-24y olds who are neither employed nor in formal education or training (NEET) reached 22% in 2021 [208].

The data we have generated using MoCA in healthy young adults with college education indicates that subjects age 21.60±5.88 have a total MoCA of 23.9±2.8, while those aged ≥31y had a MoCA average of 20.4 ± 3.4 (vs. low pollution controls 25.2 ± 2.4 p < 0.0001). This striking difference in the MoCA scores puts ̴74% of the MMC young adult middle-class population experiencing cognition deficits [203].

In the NIA-AA research framework [181], the term Alzheimer’s disease AD refers to pathological processes and in a living person is defined by biomarkers [181], thus we are in a very precarious situation: we are unable to use any of the invasive biomarkers in our young populations. Since the AT (N) system is designed to incorporate new biomarkers beyond those currently in place, there is an urgent need to have noninvasive biomarkers of P-tau and Aβ42 abnormal protein deposits across the disease continuum starting in childhood.

If we focus on AD as a continuum as per NIA-AA [181], i.e., progressive cognitive decline along progression of biomarker measures; we have APOE3 and 4 MMC children and young adults in the Alzheimer continuum from the first decade on, with a striking significant accelerated progression in young APOE4 carriers. The risk of progressing to P-tau stage 3 or higher is significant for APOE4 subjects (p = 0.0089), while the risk of progressing to Aβ phase 3 or above, is also higher for APOE4 subjects (p = 0.0035), both after adjusting age and gender [26].

MMC residents have abnormal protein deposits that define biological AD in combination with α-synuclein and TDP-43 pathology, which complicates the cognitive scenario [180,184]. Figure 5 shows the differences between the staging of P-tau and Aβ42 in APOE3 versus APOE4 MMC ≤40y old residents and the MoCA total scores [82-85].

Figure 5. The left graphic illustrates the P-tau [82-85] and the Aβ42 stages [209] for APOE3 and 4 Metropolitan Mexico City autopsy cases. The accumulation of P-tau for APOE4 carriers accelerates at the end of the second decade through the 4th, while amyloid has a slower accumulation. The risk of progressing to P-tau stage 3 or higher is significant for APOE4 subjects p = 0.0089, while the risk of progressing to Aβ phase 3 or higher is p = 0.0035, after adjusting for age and gender. The right graphic shows the average MoCA total scores for both APOE3 and 4, average values do not reach normal values (26-30) and after the middle of the 2sd decade MoCA total score goes down coinciding with the P-tau and amyloid accumulation.

Thus, essentially in MMC subjects, the abnormal protein deposits that define AD are present from infancy and the AD Continuum is seen in pediatric, teens and young adult populations. The critical periods for interventions, particularly for APOE4 carriers, are the first two decades of life. As the current research framework for AD does not incorporate air pollution parameters, there is an urgent need to include cumulative particulate matter exposures from conception and have access to PM2.5 and nanoparticle data, and their composition.

APOE4 is clearly playing a key role for AD pathogenesis (Figure 5). Neuropathological and clinical, behavioral and brain structural imagen changes start in pediatric ages.

No longer the current global estimates on numbers of people in early AD stages (prodromal and preclinical) [210] are valid without taking lifelong environmental air pollution and occupational exposures into account and in particular nanoparticles exposures that nobody measures and certainly we do not regulate in the US.

We have several proposals for the study of populations exposed to high PM2.5 and nanoparticles concentrations from intrauterine life:

1.We need to longitudinally monitor pediatric populations with non-invasive AD biomarkers, reflective of the neuropathological changes.

2.We have evidence of structural brain changes in young children and metabolic changes are readily present in APOE4 children and their parents, which offers available non-invasive (N) Neurodegeneration/neuronal injury markers. We need to implement MRI, MRS longitudinal studies contrasting children in high versus low polluted cities, matching SES, mothers IQ and educational levels, age, gestational duration, breast feeding history, birth weight, sex, maternal age, education, and socioeconomic status. Cumulative PM2.5 exposures to the complex mixture of urban ambient pollution ought to be included.

3. Based on our research findings: APOE4 carriers living in highly polluted environments should be labelled as high risk for AD, thus all studies should include APOE genotyping. A highly polluted environment can be defined as one with ambient pollution concentrations that exceed the existing air quality standards in the Organization for Economic Co-operation and Development [208].

4.We can follow longitudinally BMI, BAEPs changes, gait and equilibrium abnormalities, HbA1c, and dyslipidemia profiles in APOE4 subjects. Keeping children’s normal weights is imperative and in highly polluted cities, only physical education indoor classes should be allowed at elementary, middle, and high school levels.

5.Education efforts should be aimed for healthy lifestyle behaviors starting in infancy [211] and included in a government low-cost daycare. Access to a healthy breakfast should be available in public elementary schools along with out-of-school time (OST) programs with stimulating environments including languages, music, arts, theater, and indoor sports. OST programs must provide academic enrichment and tutorial services.

6.Education and occupation are proxies for cognitive reserve (CR), defined by Stern [212] as “differences in cognitive processes as a function of lifetime intellectual activities or other environmental factors that explain differential susceptibility to functional impairment in the presence of pathology or other neurological insult.” Given that young adults with higher college and formal education years did significantly better at cognitive testing versus less formal education years [203] and young adulthood is associated with better late-life cognition [213], efforts should be made to keep young adults in school and increase their CR. The 2021 of NEET 18-24y olds in Mexico must be down from the 22% reported [208].

7. Good balanced diets implemented in pregnant women and in infancy are key requirement in our Mexican populations. Two factors are important: i. healthy dietary behavior patterns and ii. monetary resources. In a country where we have 126 million people, including 55.7 million in poverty [214] with poor access to medical assistance and deficient education, we have an increment in chronic non-communicable diseases that as Manderson and Jewett [215] commented: they are the result of poverty and the manipulation of food markets. We certainly agreed with them. Thus, the best way to fight poverty is through the creation of local well-paid jobs, the improvement in the quality of technical programs training, access to education from elementary to college levels and the retention of students at school.

8.Purpose in life promotes resilience against brain changes already observable in middle age [216]. The work of Abellaneda-Pérez and colleagues [216] is highly recommended and emphasizes the fact that having a purposeful life implies larger functional integration of the dorsal default-mode network dDMN, which may potentially reflect greater brain reserve associated to better cognitive function. Parents and teachers could help children and teens to find purpose in their lives using a combination of experiences, education, social needs and values.

  1. The importance of APOE isoforms on the developing brain functions and the development and progression of Alzheimer’s disease is clear. We need to neuroprotect individuals at high risk and certainly APOE4 is a potential therapeutic target as commented by Ayyubova [217]. Among the 21.8 million MMC residents, we potentially have 4.36 million APOE4 subjects at early higher AD risk. These individuals are targets for neuroprotective interventions where specific pathways can be targeted in the disease process [217].
  2. The NIA-AA framework can serve as a hypothesis for testing the impact of particulate air pollution, specifically ultrafine PM and nanoparticles as the key players in the processes of oxidative stress, neuroinflammation, DNA damage, protein aggregation and misfolding, and faulty complex protein quality control. We need noninvasive biomarkers indicative of the P-tau and Aβ42 abnormal protein deposits across the disease continuum starting in childhood. The need for non-invasive biomarkers is urgent. Mexican Americans are a highly vulnerable population, severely underrepresented in research [218] and there is no support for highly trained Mexican American researchers working on AD prevention.

11.Fine particulate matter PM2.5, UFP and NPs are a serious health problem in Metropolitan Mexico City [185-189]. The problem of particle pollution is solvable. We are knowledgeable of the impact of old heavy diesel vehicles and other sources of excessive emissions and the brain effects of combustion-generated NPs [61-63]. We have the technological capability and the resources to control this pollution and protect young brains. Unfortunately, we lack the political will and aptitude to acknowledge that doing nothing will be far more costly.

All these recommendations should be implemented early, knowing cognition deficits and brain structural changes start in the first two decades of life: the critical window of opportunity to implement programs to reduce PM2.5, UFP and NPs emissions alongside with actions directed to protect children from Alzheimer’s disease and other fatal neurodegenerative diseases. APOE4 carriers are a high AD risk group.

Alzheimer’s disease evolving from childhood in polluted, anthropogenic, and industrial environments ought to be preventable.  Neuroprotection should be our goal and effective reductions of PM2.5, UFP and NPs emissions from all sources are urgently needed.Prevention ought to be at the core of the public health sector response.

Thank you for your feedback, we deeply appreciate it. Thanks again.

The paper is very technical and well-documented. However, as suggested in the comments, I strongly recommend a deep revision of English language, since in some points of the manuscript I consider difficult to understand what authors are explaining.

Authors Comment:

We had an English-speaking author to check the grammar. No money for an Editor job……….. Thank you!

Round 2

Reviewer 2 Report

The authors were asked to provide data from animal studies not to just quote the corresponding references.   If the authors provide no description of those papers, the reader would need to obtain the references and read the papers.  The same is true of the autopsies done in Mexico by the authors.  In addition to the descriptions of the studies, Tables summarizing the animal and human studies will be needed.

N/C

Author Response

R 2 REVIEWER #2

The authors were asked to provide data from animal studies not to just quote the corresponding references.   If the authors provide no description of those papers, the reader would need to obtain the references and read the papers.  The same is true of the autopsies done in Mexico by the authors.  In addition to the descriptions of the studies, Tables summarizing the animal and human studies will be needed.

AUTHORS COMMENTS

            Animals are important and we included several references (i.e.,35,54,60,62,66,45,68,76,77 etc.,) addressing important issues using experimental animals. The description is clear and concise, and some examples are included in the next paragraphs:

2.Peripheral APOE4 effects

APOE impacts extraneural peripheral tissues, and it relates to their production in hepatocytes, adipose cells, kidney and macrophages, all contributing to the peripheral APOE pool and effects [2,19, 23, 35, 42-49]. Martínez-Martínez and colleagues work is a highly recommended key reference in this section [43]. APOE polymorphic alleles directly influence plasma and brain cholesterol concentrations [43,50,51] and the liver is key for Apolipoprotein E (ApoE) -the cholesterol carrier, crucial for lipid transport and injury repair in the brain-reuptake, its release in the circulation and impaired recycling of ApoE interfering with intracellular cholesterol transport. This interference contributes to the pathophysiological lipoprotein profile observed in APOE4 carriers and impacts the pathological accumulation of triglycerides and various lipids in hepatocytes [44, 52,53].The importance of peripheral APOE in brain AD pathology and cognition was superbly demonstrated by Liu and co-workers [35] in conditional mouse models expressing human APOE 3 or 4 in liver, without brain APOE and the brain abnormal synaptic plasticity and cognition resulting from abnormal cerebrovascular function. This work showed a very important piece of the brain APOE4 puzzle: liver-expressed APOE4 exacerbates amyloid pathology in mice models [35,54].The issue is critical for humans, because in young urbanites exposed to PM pollution, including metal, metalloid, organic and inorganic toxic-carrying NPs [15,16, 26-29,38, 40]: NPs reach every tissue in the body, including the brain, the blood-brain-barrier (BBB) and liver and thus what we see as separate entities in experimental mice [35], is already in place, in humans. Liver ApoE in highly exposed NPs urbanites readily could worsen the NPs BBB damage and the brain, especially because the hepatocytes, Kupffer (KC) and endothelial sinusoidal cells are affected by NPs-induced oxidative stress, sinusoidal dilatation, Kupffer cell hyperplasia, and liver inflammation [55-57]. Arsiwala and collaborators [58] showed increased liver monocytes and KC becoming apoptotic upon the IV injection of iron NPs preparations to treat iron deficiency. The presence of nanoparticles in both hepatocytes and KC in highly exposed urbanites (personal observation Angélica González-Maciel, Rafael Reynoso-Robles, Lilian Calderón-Garcidueñas) and the acute effects upon KC upon IV Fe treatment, makes the issue very relevant to liver NPs oxidative stress and inflammation [55-58]. 

Extraordinarily relevant to the associations between cardiovascular morbidity and Alzheimer’s disease is the report of Habenicht and collaborators [59] associating a C1q-ApoE complex in intimal arterial lesions and human amyloid plaques. As described by the authors, the association of C1q- an initiating and controlling protein of the classical complement cascade-, key in acute and chronic inflammatory responses and secreted by myeloid cells (including Kupffer cells). To initiate the CCC cascade, C1q must be activated by molecules as varied as oxidized lipids, amyloid fibrils, and immune complexes and ApoE mute towards inactive C1q binds at high affinity to its activated form. Habenicht and collaborators C1q-ApoE complex in intimal locations illustrates the APOE peripheral and brain detrimental effects [59].

The work of Yin and collaborators [60] showing the total hepatocyte inhibition of mitochondrial, and ATP-linked oxygen consumption rate-indicative of mitochondrial dysfunction-upon exposure to diesel NPs in APOE knockout mice illustrates a key diesel effect: mitochondrial dysfunction in both liver and brain. Yin and co-workers’ paper is of practical importance: diesel NPs are common pollutants in urban areas and occupational environments and NPs <100 nm readily found in the exhaust, are precisely the particles’ size capable of crossing all biological barriers and using cation channels such as Transient Receptor Potential (TRP) proteins involved in the regulation of intracellular biochemical signaling processes and cellular electrical excitability [61-64,27-29].

The phenotypic spectrum of non-alcoholic fatty liver disease (NAFLD) and ApoE was described by Meroni et al., [65] in a 40y old female with an early NAFLD and severe hypertriglyceridemia. Liver biopsy showed the extensive mitochondrial architecture abnormalities; thus, liver mitochondrial abnormalities are associated to genes involved in lipid metabolism and missense mutations involved in mitochondrial dysfunction. Genetic factors such as APOE and others [65] are synergistic with metabolic associated fatty liver disease, medications [43, 46,51, 55,56] and environmental sustained and seemingly harmless nanoparticles, i.e., the case of silica, magnificently illustrated in the work of Abulikemu et al [45].The respiratory exposure of silica NPs induces hepatotoxicity, the authors [45] focused on the role of SiNPs in the pathogenesis and progression of NAFLD and demonstrated a significant aggravation of hepatic steatosis, inflammation and collagen deposition in ApoE-/- mice, along with high concentrations of ALT, AST and LDH levels associated with liver damage [45]. Moreover, common NPs path mechanisms are seen in liver, heart and brain NPs damage, including endoplasmic reticulum (ER) stress [66- 71].

The complexity of APOE4 peripheral effects certainly also impacts body mass index (BMI), cardiovascular disease (CVD) risk and thus later brain effects [72,73]. Ozen et al., showed higher fasting blood lipids and higher CVD risk in APOE4 carriers with body composition and diet playing an important role [72]. There is no doubt CVD risk goes hand in hand with brain aging, thus the relationship with APOE4 becomes crucial when evaluating brain effects as in Subramaniapillai et al., work [74]. The authors evaluated 21,308 UK Biobank volunteers and established the association between white matter brain age gap (BAG) and BMI, waist to hip ratio (WHR), body fat percentage (BF%), and APOE4 status. Interestingly, older females[66-81y] with greater BF% had lower BAG, while earlier menopause transition was associated with higher BAG.

APOE peripheral effects have direct or indirect detrimental impact upon the brain.

  1. APOE4 brain effects

APOE4 is a complex risk factor for AD and other neurodegenerative diseases, with multiple, heterogenous and overlapping action mechanisms and molecular interactions [3, 5, 6, 10, 14, 30, 35, 46,75-80]. Stuchell-Brereton et al., [75] work has shown that APOE4 is far more disordered and extended than previously described and retains conformational heterogeneity after binding lipids, which helps to understand the differences between the ɛ4 allele and protective variants of the protein. While there is not much controversy in stating APOE4 increases the risk of AD by driving earlier and more abundant amyloid pathology in the brains of APOE4 carriers [30], the work of Lazar et al., [76] in mice expressing the human APOE4 vs. APOE3 isoform shows APOE4 carriers had altered cholesterol turnover, ratio imbalances of specific classes of phospholipids, low phosphatidylethanolamines bearing polyunsaturated fatty acids and an elevation in monounsaturated fatty acids. More importantly, these changes in lipid homeostasis were related to increased production of Aβ peptides, and higher levels of tau and phosphorylated tau in primary neuronal cultures [76]. Tcw et al., [77] investigated the effects of APOE4 on neural cells and isogenic human induced pluripotent stem cells, as well as in human autopsies and in APOE targeted replacement mice. APOE4 driven metabolic dysregulation of astrocytes and microglia are major findings in their work. The significant increases in cholesterol synthesis in APOE4 astrocytes already having lysosomal cholesterol sequestration illustrates the ɛ4 impact associated with matrisome dysregulation. The astrocyte pathology is key in APOE4 carriers and certainly contributes to the complexity of the abnormal relationship with neurons and the neurovascular unit [78,79].

…………………….

APOE4 alters neuronal insulin signaling in human APOE-targeted replacement (TR) mice and in APOE4-TR mice feed a high-fat diet, accelerates these effects at middle age [131]. APOE4 interacts with the insulin receptor and impairs its trafficking by trapping it in endosomes, leading to impaired insulin signaling and insulin-stimulated mitochondrial respiration and glycolysis [131]. The disruption in the fatty acid regulation alters both neurons and astrocytes and decreases FA astrocytic transport [132].

Given that the focus of the paper is human reports, we have added in the abstract the word human

Abstract 

This Review emphasizes the impact of APOE4-the most significant genetic risk factor for Alzheimer's disease (AD)- upon peripheral and neural effects starting in childhood. We discuss major mechanistic players associated with APOE alleles’ effects in humans, in order to understand their impact from conception and through all life stages and the importance of detrimental, synergistic environmental exposures.

The same is true of the autopsies done in Mexico by the authors.  In addition to the descriptions of the studies,

In the second review you asked for a description of the autopsies done in Mexico.

Here it is:

Forensic autopsies in MMC children and young adults exhibit the neuropathology hallmarks of Alzheimer and Parkinson’s diseases and TDP-43 pathology [26,27,39,40] and metal and metalloid NPs are seen in intracellular locations in association with damaged mitochondria, endoplasmic reticulum, Golgi apparatus, heterochromatin, etc., involving the brain and heart tissues [26-28, 71,89,100,101]. We have also documented early and progressive neurovascular unit damage and key organelle ultrastructural endothelial pathology associated with NPs [38, 42, 89].Breakdown of the BBB can be documented in infants and the NPs extensively damaged both endothelial cells and their tight junctions [26,27,38,42].Interestingly, MMC organic PM2.5 components include lipopolysaccharides (LPS), thus neuroinflammation and BBB remodeling-associated effect as described by Erickson et al., [190] in LPS-treated mice is seen clearly in MMC residents.

AD neuropathology markers present in 202 of 203 MMC autopsies in subjects ≤40y old, including 44 children are illustrated in Figure 2: hyperphosphorylated tau and beta amyloid per decade [26]. These subjects had no extra neural light microscopy pathology.

Figure 2. Alzheimer’s disease staging using P-tau (Htau) and Aβ42 in 203 autopsies [26] average age 25.4±9.2y.  202/203 had AD pathology, including the youngest subject, an 11-month-old baby.

 All children in the first decade of life had P-tau pre-tangle stages and by the 2sd decade we documented neurofibrillary (NFT) tangles I-V [26, 191]. Subjects in the 4th decade were in NFT I-V stages and we could no longer document pre-tangle stages. In contrast, Aβ progressed slowly and was kept in the early phases. Remarkably, in our autopsy studies [26], APOE 4 carriers had higher NFTs Braak stages and the highest risk for suicide was associated with lower cumulative exposures to PM2.5 versus APOE3 carriers (Figure 3).

Figure 3. Cumulative suicide risk probability between MMC APOE3 and 4 subjects as a function of cumulative PM2.5 lifelong exposures-including pregnancy-, after adjusting for age. Cox’s proportional hazard model was used to estimate the underlying survival function and the individual’s age was used as a predictor [26].

We have described the remarkable AD, PD and TDP-43 pathology overlap in young MMC residents and the fact that is very similar to the mixed protein pathologies in elderly patients with AD, frontotemporal lobar dementia (FTLD), Lewy body disease (LBD), Parkinson’s disease (PD), amyotrophic lateral sclerosis (ALS), cerebral amyloid angiopathy (CAA), and in younger than 60y AD demented patients [180,184, 192-196].  Hyperphosphorylated tau is the major aberrant protein in MMC young residents and P-tau in substantia nigra and lack of nuclear TDP-43 in cortical motor neurons, motor neurons for cranial nerves III, V and XII and cervical motor neurons in teens and young adults were common findings [26-28].  Figure 4 shows the aberrant neural protein overlap in MMC 186 young residents compared [27] to elderly subjects in the work of 375 autopsies done by Karanth et al [184].

Figure 4. Comparison in aberrant neural proteins between the young MMC 186 autopsy cohort [27] and Karanth et al., [184] 375 autopsies with an average age of 86.9±8.04y including subjects with normal cognition, mild cognitive impairment (MCI), impaired (but not MCI), and dementia.

Localization of highly toxic Fe, Ti, Hg, W, Al and Zn spherical and acicular NPs in the locus coeruleus (LC), neural and vascular mitochondria, endoplasmic reticulum, Golgi, neuromelanin, heterochromatin and nuclear pore complexes along with early and progressive neurovascular damage is of upmost importance for MMC young residents [89] in view of Mercan and Heneka work [197]. Mercan and Heneka emphasized LC neuronal loss as one of the earliest indicators of neurodegeneration in AD. The authors commented about how LC degeneration results in decreased noradrenalin levels, increased neuroinflammation, enhanced amyloid and tau burden and cognition impairment and decrease long-term synaptic plasticity. Alterations in the locus coeruleus-noradrenaline system are important contributors to AD progression and indeed severe loss of LC cells and subcellular damage associated to NPs are present in young children in MMC [197,27,89]. Remarkably, FeNPs 4 ± 1 nm and Hg NPs 8 ± 2 nm were seen predominantly in the LC and substantia nigra (SN) [27,89]. NPs damage to the substantia nigrae and cerebellum are outstanding and translate in autonomic dysfunction, gait and balance alterations and cerebellar MRI atrophy in young MMC residents [89,90,101, 41,198].

Nanoparticles were also present at all placental stages, including 8-15week placentas [29]. NPs were documented in maternal red blood cells (RBC), early syncytiotrophoblast, Hofbauer cells, and fetal endothelium (ECs). Fetal ECs displayed caveolar NP activity and widespread erythroblast-loaded NPs contact. Erythroblasts are the main carriers of NPs to the developing brain and primitive neural cells showed nuclear, organelle, and cytoplasmic Fe, Ti, and Al alloys, Hg, Cu, Ca, Sn, and Si NPs in both singles and conglomerates. Combustion-derived NPs as well as industrial NPs are documented in early fetal brains [29].

MMC residents have abnormal protein deposits that define biological AD in combination with α-synuclein and TDP-43 pathology, which complicates the cognitive scenario [180,184]. Figure 5 shows the differences between the staging of P-tau and Aβ42 in APOE3 versus APOE4 MMC ≤40y old residents and the MoCA total scores [82-85].

Figure 5. The left graphic illustrates the P-tau [82-85] and the Aβ42 stages [212] for APOE3 and 4 Metropolitan Mexico City autopsy cases. The accumulation of P-tau for APOE4 carriers accelerates at the end of the second decade through the 4th, while amyloid has a slower accumulation. The risk of progressing to P-tau stage 3 or higher is significant for APOE4 subjects p = 0.0089, while the risk of progressing to Aβ phase 3 or higher is p = 0.0035, after adjusting for age and gender. The right graphic shows the average MoCA total scores for both APOE3 and 4; average values do not reach normal values (26-30) and after the middle of the 2sd decade MoCA total score goes down coinciding with the P-tau and amyloid accumulation.

Thus, essentially in MMC subjects, the abnormal protein deposits that define AD are present from infancy and the AD Continuum is seen in pediatric, teens and young adult populations. The critical periods for interventions, particularly for APOE4 carriers, are the first two decades of life. As the current research framework for AD does not incorporate air pollution parameters, there is an urgent need to include cumulative particulate matter exposures from conception and have access to PM2.5 and nanoparticle data, and their composition.

APOE4 is clearly playing a key role for AD pathogenesis (Figure 5). Neuropathological and clinical, behavioral and brain structural imagen changes start in pediatric ages.

We hope that our extensive information and of course, the fact the reader can access the papers, will be what the reader needs when reading a review.

Thanks so much for your input.

Reviewer 3 Report

Dear authors, thank you for the great work in so few time. The paper has been largely improved. I underline some points in the manuscript that, in my opinion, should be further ameliorated.

1) The title has been shortened and it’s more clear. However, I suggest to further reduce. Moreover, I believe that the two major points chosen (acceleration of AD process and risk of suicide) are two important issues, but too strongly reported in the title. You can change as “APOE4 accelerates the pathological processes related to AD in carriers…”. The last sentence is a purpose, but I would not report in the title.

2) Abstract: I still believe that the risk of suicide for subjects APOE4 carriers should be further discussed and clarified, both in the abstract and in the manuscript. Is there a direct link between risk of suicide and the risk to reach Braak Stage V? I understand that APOE4 is related to high risk if associated with lower PM2.5 exposure than APOE3, is correct? I suggest to further discuss.

3) Line 370. Citation 165 may be redundant at the end of the sentence, in which authors state to agree with Chang and colleagues.

4) The NIA-AA framework presents some unmet needs, as correctly stated by authors. However, I cannot well undertand in what sense authors report that NIA-AA framework may serve to test the hypothesis of the impact of pollution on molecular drivers for AD. I suggest to further clarify this point, especially in the perspective of a new way of facing with AD.

None

Author Response

Dear authors, thank you for the great work in so few time. The paper has been largely improved. I underline some points in the manuscript that, in my opinion, should be further ameliorated.

  • The title has been shortened and it’s more clear. However, I suggest to further reduce. Moreover, I believe that the two major points chosen (acceleration of AD process and risk of suicide) are two important issues, but too strongly reported in the title. You can change as “APOE4 accelerates the pathological processes related to AD in carriers…”. The last sentence is a purpose, but I would not report in the title.

AUTHORS RESPONSE

APOE peripheral and brain impact.

APOE4 carriers accelerate their Alzheimer Continuum and have a high risk of suicide in highly polluted cities.

Control of anthropogenic and industrial nanoparticle emissions and early neuroprotection are urgent.

The data in Mexico City fully support the relationship between APOE4 and suicide and in fact, private physicians familiar with our publications are now asking for brain MRI to check the possibility of brain atrophy at younger ages and neuroradiologists are picking up such cases.

In addition, recent literature J Affect Disord 2022 Dec 15;319:202-212 doi: 10.1016/j.jad.2022.09.046 in Italy discusses suicidal ideation and biopsychosocial predictors in older age and IL-6 followed by TNF-α, APOE ε4 allele presence, CRP and high-density lipoprotein cholesterol contributed most to suicidal ideation.

In the USA, Alzheimer's disease (AD) and AD-related dementias (ADRDs) are at a higher risk of suicidal behaviors given intersecting risk factors.

Alipour-Haris G, Armstrong MJ, Sullivan JL, Survadevara U, Rouhizadeh M, Brown JD. Suicidal Ideation and Suicide-Attempt-Related Hospitalizations among People with Alzheimer's Disease (AD) and AD-Related Dementias in the United States during 2016-2018 J Clin Med2022 Feb 11;11(4):943.  doi: 10.3390/jcm11040943.

Interestingly, in the period 2016-2018, there were 12,538 hospitalizations related to suicidal behaviors for people with AD/ADRDs. The overall prevalence of suicidal-behavior-related hospitalizations was lowest for AD (0.8%) and highest for frontotemporal dementia (2.6%). The average age? 73.26±, so we are able in Mexico City to pick up the higher risk of suicide almost 50 y earlier. Somebody has to know that! Every urban center has a Forensic laboratory at hand, people die every day, just do a staging AD,PD, TDP-43 neuropath examination and genotype subjects, you will find out immediately.

I live in MONTANA where we have a very high risk of suicide versus the rest of the USA. Lots of people have wood fireplaces (awesome source of nanoparticles), forest fires are common in the Summer and tobacco smokers are numerous (plus marijuana recently).

There has been a 30% increase in the number of suicides in the United States since 1998. (CDC, 2018) In 2020 there were 45,979 suicides in the U.S. (129 suicides per day; 1 suicide every 11 minutes).

Thus, a key issue that likely determines APOE4 is a factor in suicide is the fact that in high polluted scenarios E4 carriers accelerate their P-tau stages, as we have described in Mexico City. And please keep in mind, PM2.5 is not a good proxy for PM pollution, so comparisons between cities and countries are not accurate. I can only speak for Mexico City residents and here, you have an APOE4 you have AD.

  • Abstract: I still believe that the risk of suicide for subjects APOE4 carriers should be further discussed and clarified, both in the abstract and in the manuscript. Is there a direct link between risk of suicide and the risk to reach Braak Stage V? I understand that APOE4 is related to high risk if associated with lower PM2.5 exposure than APOE3, is correct? I suggest to further discuss.

Thus, in response to your Comment here is the modified pertinent paragraph and added 3 references:

The issue of young APOE4 males at high risk for suicide over their APOE3 counterparts [26] should be emphasized. First, males are more likely to comitte suicide using weapons or other means, with high chances of death outcomes(i.e., hanging) and thus the forensic data shows mostly males, their key finding was their APOE4 status and how at the same cumulative PM2.5 exposures as their APOE3 counterparts, they strikingly accelerated P-tau stages [26]. This is not a suprising finding, AD and AD-related dementias (ADRDs) patients are at a higher risk of suicidal behaviors as shown by Alipour-Haris et al., [207] in 12,538 US hospitalizations (2016-2018) related to suicidal behaviors and in suicidal ideators with late-life depression in whom TNF-α, APOE ε4 allele presence, CRP and high-density lipoprotein cholesterol contributed most to suicidal ideation [208].The work of Abramova et al., [209] is extraordinarily important for megacities’ exposed populations. They genotyped 146 healthy controls and 112 dementia patients for APOE rs429358rs7412 and genes associated to suicide: an association with dementia of rs4918918 (SORBS1) and rs10903034 (IFNLR1) previously associated with suicide was established and they confirmed the association of APOEε4 with dementia.

  • Line 370. Citation 165 may be redundant at the end of the sentence, in which authors state to agree with Chang and colleagues.

AUTHORS COMMENTS

DONE!

The authors concluded defining APOE ε polymorphisms in young children may provide the earliest indicators for individuals who might benefit from early interventions or preventive measures for future brain injuries and dementia [165]; we fully agreed with them.

  • The NIA-AA framework presents some unmet needs, as correctly stated by authors. However, I cannot well undertand in what sense authors report that NIA-AA framework may serve to test the hypothesis of the impact of pollution on molecular drivers for AD. I suggest to further clarify this point, especially in the perspective of a new way of facing with AD.

AUTHORS COMMENTS

The NIA-AA framework clearly states the presence of P-tau and Aβ In the NIA-AA research framework [181], the term Alzheimer’s disease AD refers to pathological processes and in a living person is defined by biomarkers [181.

The focus of the NIA-AA is the elderly population, but they are completely ignoring the highly exposed young subjects exhibiting already the presence of P-tau and Aβ. There is a good reason for that, preventive medicine is not profitable, thus why try to protect the youngest, if you will be ready to give them an expensive (useless) pill when their brain weights 900g? We need to add UFPM and NPs measurements to the research framework.

We have 21.8 million people in an amazingly effective exposure chamber, we know they are developing AD, PD and TDP-43 pathology, we have a remarkable multidisciplinary team ready to go and lots of volunteers, so what else they need to support real life research?

 Thanks so much for your input, it is a pleasure to answer coherent questions from reviewers willing to help the authors to improve their work.

Round 3

Reviewer 2 Report

None

Author Response

We thank you!!!!!!!!!!!!!!!!!!!!!!

Calderon-Garciduenas et al., 

Reviewer 3 Report

Dear authors, I appreciate very much your efforts to improve the manuscript. I believe that each contribution on this relevant field deserves to be considered and deepened.  I suggested to authors to further shorten the title.

English requires only a minor revision.

Author Response

Thank you!!!!!!!!!!!!!!!

Shorten title:

APOE peripheral and brain impact: APOE 4 carriers accelerate their Alzheimer Continuum and have a high risk of suicide in PM2.5 polluted cities